# Predicting glycan structure from tandem mass spectrometry via deep learning

**James Urban[1,2], Chunsheng Jin[3], Kristina A. Thomsson[3], Niclas G. Karlsson ⓘ [4], Callum M. Ives ⓘ [5], Elisa Fadda[6] & Daniel Bojar ⓘ [1,2]✉**

Glycans constitute the most complicated post-translational modification, modulating protein activity in health and disease. However, structural annotation from tandem mass spectrometry (MS/MS) data is a bottleneck in glycomics, preventing high-throughput endeavors and relegating glycomics to a few experts. Trained on a newly curated set of 500,000 annotated MS/MS spectra, here we present CandyCrunch, a dilated residual neural network predicting glycan structure from raw liquid chromatography–MS/MS data in seconds (top-1 accuracy: 90.3%). We developed an open-access Python-based workflow of raw data conversion and prediction, followed by automated curation and fragment annotation, with predictions recapitulating and extending expert annotation. We demonstrate that this can be used for de novo annotation, diagnostic fragment identification and high-throughput glycomics. For maximum impact, this entire pipeline is tightly interlaced with our glycowork platform and can be easily tested at https://colab.research.google.com/github/BojarLab/CandyCrunch/blob/main/CandyCrunch.ipynb. We envision CandyCrunch to democratize structural glycomics and the elucidation of biological roles of glycans.

As the most abundant post-translational modification, glycans are frequently dysregulated and mechanistically involved in diseases ranging from cancer[1] to metabolic disorders[2]. The exact structure of complex carbohydrates is often key in mediating their function[3], such as sialic acid only facilitating influenza infection in a particular linkage orientation[4]. From biomarkers to mechanistic understanding[1,2], structural resolution thus is relevant for integrating and using glycan information for biomedical gains. In the context of systems biology, glycans are routinely measured via mass spectrometry (MS)-based glycomics[5], providing insights into which structures or substructures are dysregulated, which can be further analyzed with various methods[6,7].

Currently, structural determination of glycans is, at best, semi-manual and proceeds structure by structure[8]. Since different glycan structures can result in the same mass, structural isomers are

routinely separated via liquid chromatography (LC)[9], followed by fragmentation into smaller substructures by MS, conceptually akin to shotgun sequencing. Current in-depth workflows are hard to parallelize, with a general trade-off between resolution and scale[10]. All this has relegated structural glycomics to a few experts, inaccessible to most life science researchers.

Extensive work by Harvey and others[8,11,12] has demonstrated that, in principle, most substructures[13], linkages[14] and monosaccharides[15] have diagnostic fragments or intensity ratios. Using this fine structural information that is contained within MS/MS spectra, along with basic biosynthetic assumptions, it is thus frequently possible to achieve high-resolution annotations of native glycans. In practice, however, annotation is often restricted to essentially topological assignments, not least due to time-constraints. Nuances of diagnostic indicators are challenging for humans to decrypt manually or encode

[1]Department of Chemistry and Molecular Biology, University of Gothenburg, Gothenburg, Sweden. [2]Wallenberg Centre for Molecular and Translational Medicine, University of Gothenburg, Gothenburg, Sweden. [3]Proteomics Core Facility at Sahlgrenska Academy, University of Gothenburg, Gothenburg, Sweden. [4]Section of Pharmacy, Department of Life Sciences and Health, Faculty of Health Sciences, Oslo Metropolitan University, Oslo, Norway. [5]Department of Chemistry and Hamilton Institute, Maynooth University, Maynooth, Ireland. [6]School of Biological Sciences, University of Southampton, Southampton, UK. ✉e-mail: daniel.bojar@gu.se

programmatically, especially at scale and accommodating diverse experimental setups, as each linkage and monosaccharide can be affected by its sequence context[16]. This combinatorial explosion, combined with rich data, is promising for scalable artificial intelligence (AI) approaches which can learn complex mapping functions, as recently demonstrated by endeavors such as AlphaFold2 (ref. 17).

So far, computational attempts to automate MS-based glycomics[18–23] did not engage with deep learning. Rather, they relied on various search methods, to search for either possible topologies given a precursor ion mass or suitable reference spectra, loose constraints that may yield unphysiological predictions. Their primary limitations are scale and annotation resolution, ranging from composition to glycan topology. Neither linkage type nor monosaccharide stereoisomers are commonly resolved during this algorithmic sequencing. Additional hurdles to their wider adoption include poor generalizability, as none of them employ a rigorous train–test mentality, a standard practice in machine learning to evaluate methods on held-out data to prevent overfitting. Many tools were designed for very specific problems and were often tested on few spectra[18–20], precluding their usage in many experimental setups.

Recent efforts in related fields, particularly in proteomics[24,25], have employed scalable deep learning strategies in MS analysis. Proteomics has partially similar challenges to glycomics, for example, precursor structure elucidation given fragment ions. We thus posit that the translation of analogous methods to structural glycomics, combined with domain-inspired additions such as biosynthetic constraints and building on the accumulated work of many years of glycomics analysts, could be a major leap forward for the field and the usage of glycomics in the broader life sciences.

We present a scalable and accurate workflow for predicting glycan structure from liquid chromatography with tandem mass spectrometry (LC–MS/MS) data, centered on our deep learning model, CandyCrunch. Using a large-scale, curated set of tandem spectra from diverse experimental setups, CandyCrunch predicts glycan structure with high accuracy (~90%), outperforms existing methods on this task and matches/extends expert annotations on unseen data. This is facilitated by various domain-specific advancements, for example, considering glycan structure similarity in the loss function. We embedded this into a downstream workflow converting predictions into interpretable results, further reducing false positive rates, and estimating relative abundances; all in seconds. This workflow includes CandyCrumbs, a comprehensive MS/MS fragment annotation plug-in we developed here. We used this to uncover diagnostic fragments and more complex fragmentation behavior at scale, underpinned by molecular dynamics simulations. Finally, we annotate novel glycomes, analyze biosynthetic constraints at scale and demonstrate that our pipeline can be used in high-throughput glycomics. Our methods are accessible within a Python package (https://github.com/BojarLab/CandyCrunch), a free-standing Google Colab notebook at https://colab.research.google.com/github/BojarLab/CandyCrunch/blob/main/CandyCrunch.ipynb and a command line interface available via our Python package (further usage description at https://github.com/BojarLab/CandyCrunch).

## Results

### CandyCrunch predicts glycan structure via domain knowledge

Reasoning that the fragmentation patterns and propensities (that is, intensity ratios) in MS/MS are predictive of glycan structure—a relationship that is used by human experts in annotation—we set out to learn this association via machine learning. For this, we collected and curated an unprecedentedly large set of annotated LC–MS/MS spectra that derive from glycans (Fig. 1a,b and Methods). We envision that, even beyond our efforts here, this dataset will be a valuable resource for data-driven approaches in glycomics. Crucially, this dataset aims to provide a representative view over current glycomics data, with a total of nearly 500,000 labeled MS/MS spectra from >2,000 glycomics experiments, encompassing all major eukaryotic glycan classes (N-linked, O-linked, glycosphingolipid, milk oligosaccharides) and the most common experimental setups for glycomics. The exact composition of this dataset, broken down by glycan classes and experimental parameters, can be found in Supplementary Table 1. To avoid overrepresenting some classes (for example, core 1 O-glycan), we then limited each class to a maximum of 1,000 spectra in the independent test set (see Methods for details) and used the remaining ~450,000 spectra to train our model on the most likely glycans in a multiclass classification setup (Methods).

This resulted in our dilated residual neural network, CandyCrunch, a model architecture suited to MS data[25]. Since experimental parameters such as the ion mode drastically change fragmentation patterns, it uses the MS/MS spectrum, retention time, precursor ion m/z and experimental parameters (for example, LC type, ion mode and so on) as input and predicts glycan rankings as its output (Fig. 1c), using information from these different sources of input which are only partly redundant (Supplementary Tables 2 and 3). We note that we neither claim, nor sought to obtain, the most frugal model for this task, but rather the most performant and flexible, without noticeable hardware limitations (CandyCrunch can be readily used on a typical laptop). Our current binning strategy lowers the effective resolution of the mass spectrometer. Yet we note that, for the moment, analyzing the data at higher resolution, more closely approximating the true instrument resolution, does not give rise to higher accuracy (Supplementary Table 4), as most fragments are uniquely specified by our current binning method (Supplementary Fig. 1 and Methods). Further, capturing minute mass differences such as between $CH_4$ and O (0.036 Da) would require impractically fine binning and is invalidated by the large proportion of low-resolution data in our training dataset. Available options to run CandyCrunch are shown in Supplementary Fig. 2. The model is part of a pipeline applied to a raw file (for example, .mzML or .mzXML files), which groups predictions based on mass and retention isomers and further curates predictions with, for example, diagnostic ions (Fig. 1d). We confirmed that this grouping procedure even succeeded in the case of retention time overlaps between peaks (Supplementary Fig. 3), although we caution that biological samples may contain more closely co-eluting structures that are not disambiguated by human annotators in the data used to train CandyCrunch.

If precursor ion intensities are available in the raw file, this pipeline can also estimate relative abundances. These abundances correlate well

**Fig. 1 | Predicting glycan structure via deep learning. a,b**, Overview of the curated dataset of glycomics LC–MS/MS by glycan class (**a**) and source (**b**). Diagonal bars indicate positive ion mode data. The numbers correspond to spectra with annotations. **c**, Schematic view of CandyCrunch model architecture. **d**, Pipeline of curating glycan predictions from raw file to final output table. **e**, Evaluating top-1 accuracy on the independent test set (Methods; see ref. 43) across different levels of resolution. **f**, Learned representations of all spectra in the test set are shown via t-distributed stochastic neighbor embedding (t-SNE), colored by glycan class. Examples are illustrated with their glycan structures. **g**, Excerpt from an example prediction output using our Colab notebook on the file JC_171002Y1.mzML (ref. 44). **h**, Proportional Venn diagram of the comparison of CandyCrunch and Glycoforest on the raw file JC_131210PMpx5.mzML (ref. 18), not used for training CandyCrunch but used for developing Glycoforest. Shown are topologies (Glycoforest does not output full structures) matching those of the human annotator for each model (see Supplementary Fig. 11 for detailed comparison). All masses shown are from reduced glycans. Glycans here and in the entire paper are drawn using GlycoDraw[45] according to the Symbol Nomenclature for Glycans (SNFG). Conv, convolutional layer; d, dilation; MO/GSL, milk oligosaccharides/glycosphingolipids; PGC, porous graphitized carbon chromatography; RT, retention time.

with those gained by LC peak area integration (Supplementary Fig. 4), a state-of-the-art approach for estimating relative abundances. We caution that overlapping isomer peaks may lead to moderate

uncertainties in their quantification. Overall, CandyCrunch is highly performant, with an accuracy of ~90% of the top-ranked structure prediction in the independent test set (Fig. 1e), performing comparably

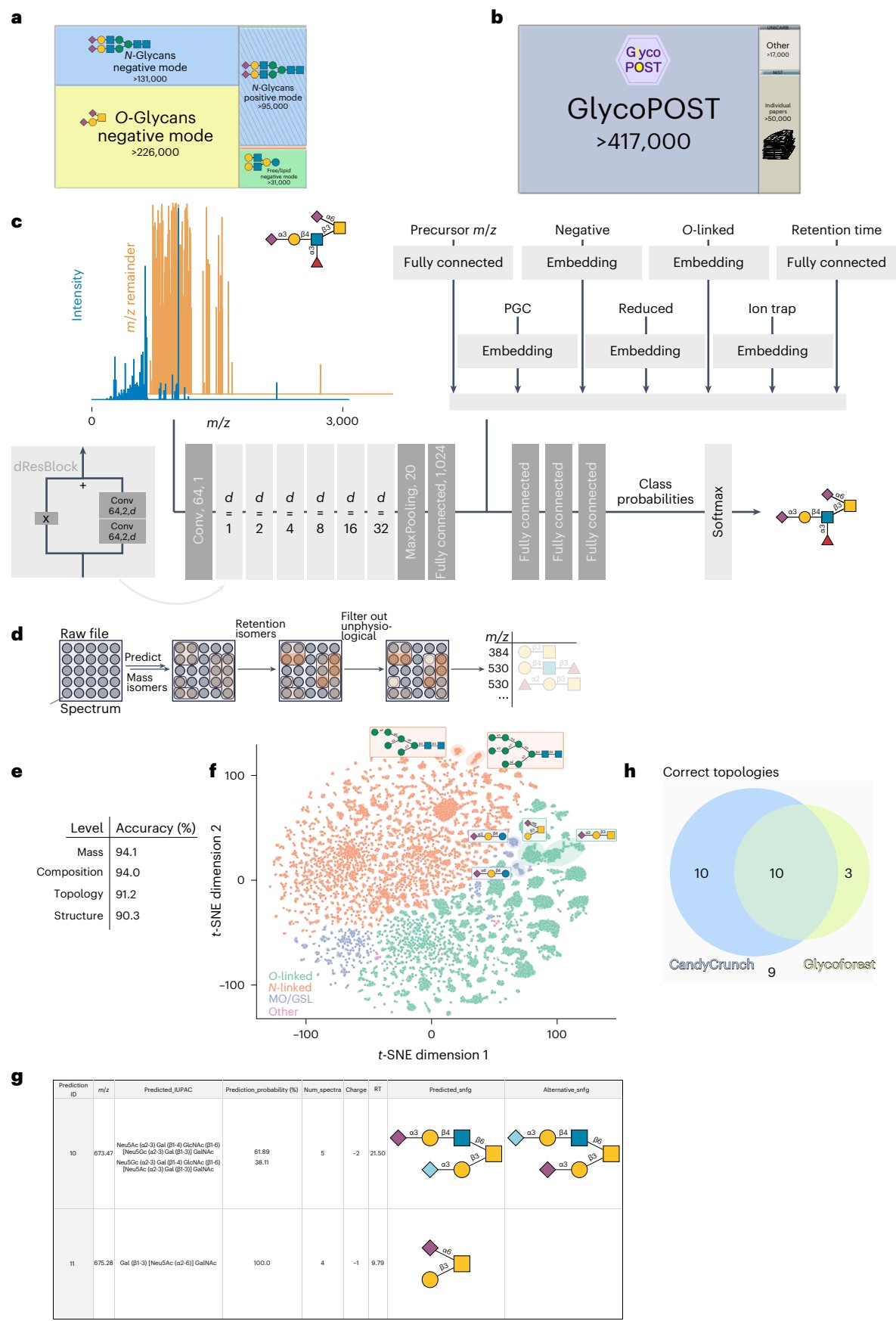

across glycans (Supplementary Fig. 5a,b) and across different MS setups, glycan classes and derivatized glycans (Supplementary Tables 2, 5 and 6), albeit with lower performance on data-poorer categories such as permethylated glycans. We also note that any evaluation is partly confounded by different annotation qualities, which may be, for instance, substantiated by exoglycosidase treatment in some cases but not in others, resulting in more ambiguous 'ground truths'. Consequently, higher-quality data further improve performance, reaching up to ~95% accuracy currently (Supplementary Table 6). Custom loss functions estimating structural distance to the ground truth, and many more domain knowledge-inspired modifications (Methods), ensure that even erroneous predictions are structurally close to the correct solution. We quantified this statement by analyzing that structures with more shared motifs need fewer of their 'own' spectra to reach high prediction accuracy, indicating effective cross-training (Supplementary Fig. 6). Our approach also includes incompletely resolved structures, so that prediction uncertainty can be meaningfully conveyed via, for instance, missing linkage information (indicated by a higher topology accuracy than structure accuracy; Fig. 1e). Further, the prediction score is a meaningful indicator of confidence and, when comparing top-1 predictions of the same structures, is higher for correct predictions (Supplementary Table 7).

Learned representations of spectra by CandyCrunch cluster by glycan sequence and glycan class (Fig. 1f), demonstrating that the model has learned to accommodate experimental variability. Further, structurally related glycans, even within the same class, tend to cluster together in the learned representation space. This can be quantified by comparing the cosine distance of learned representations of pairs of glycans with their structural distance, revealing that the co-clustering described by the representations is indeed suggestive of structural relatedness of glycans (two-sided Mantel test of correlating the two resulting cosine distance matrices; $P < 0.001$), already alluded to via Supplementary Fig. 6 above.

In framing CandyCrunch as a multiclass classification problem (that is, ranking the likelihood of pre-defined glycans), we minimized the chance for unphysiological glycans in the output, a very real possibility otherwise, given the sparsity of real glycan sequences among possible sequences[26]. However, this made zero-shot predictions—predicting a glycan sequence that was absent from our training set—conceptually infeasible. As repositories such as GlycoPOST do not catalog all physiological glycans, and glycomics studies, such as mucin-type O-glycomics[27] or milk glycomics[28], routinely discover new structures, we set out to augment our pipeline to allow for, limited, zero-shot prediction outside our 3,391 defined glycans.

Reasoning that glycans in a biological sample tend to be biosynthetically related, that is, contain precursors/intermediates of larger biosynthetic pathways, we turned to our recently developed method of constructing glycan biosynthetic networks[7]. Applying this method to a typical CandyCrunch output (Supplementary Fig. 7) revealed the existence of necessary intermediate structures that were absent from our predictions but would explain spectra without a valid prediction. We thus added this routine as an optional step in our inference workflow, to facilitate a certain subset of physiological zero-shot predictions, which we support empirically (Supplementary Fig. 8). We caution that this additional workflow step is only expected to add value if mixtures

of related glycans, such as in cells, blood or tissue, are analyzed, not purified synthetic structures.

CandyCrunch is fundamentally database-independent but can be further enhanced by methods leveraging databases, such as defined within glycowork[29], to augment predictions downstream. By carefully selecting a suitable subset of reference structures (for example, by taxonomy, glycan class or tissue), matches for unexplainable spectra could be proposed. These potential matches were then cross-checked for diagnostic ions as well as ranked by biosynthetic compatibility with true predictions. This, again, allowed for a certain subset of zero-shot predictions. It should be noted that this procedure still balanced the theoretical constraint of physiological glycans with the reality of encountering novel structures in biological samples. Our final inference workflow then also contained this latter expansion, resulting in a ranked prediction output that can be further investigated by the researcher (Fig. 1g). It should be noted, however, that the default in the provided notebook is not to run zero-shot inference, as this requires much more expert review than our regular model-based inference. We also developed a workflow for batches of samples, which accommodates shifts in retention time by grouping peaks across samples, resulting in improved predictions (Supplementary Fig. 9).

Next, we compared CandyCrunch with alternative approaches to this problem. As a preface, we should note that no current approach combines CandyCrunch's advantages of scale, generalizability, performance and its flexibility in usage (Supplementary Fig. 10a). Further, most methods are maintained for only the briefest of periods and are no longer realistically accessible. Thus, we had to effectively constrain ourselves to compare CandyCrunch on individual raw files that were specifically used to build these alternative approaches, while we excluded them during training. Still, in direct comparison with state-of-the-art methods such as Glycoforest[18] on challenging fish mucin glycans, CandyCrunch demonstrated a greater overlap with manual expert annotations (Fig. 1h; 62.5% versus Glycoforest's 40.6%) and a substantially higher structural resolution (Supplementary Fig. 11). In addition, by tethering CandyCrunch and the below-mentioned CandyCrumbs to our glycowork ecosystem[29] and by providing everything open-source, we substantially increase the chances for the long-term viability of our presented methods.

Applied to fully unseen datasets, CandyCrunch routinely achieved high performance (Supplementary Table 8; topology: 75.3% top-1 accuracy, structure: 72.4% top-1 accuracy) and potentially can extend expert annotations by correctly capturing additional structures and isomers (Supplementary Fig. 12). The additional predictions in this sample partly even stemmed from remnant glycans from the previous sample, showcasing the exceptional sensitivity of our model. We would like to highlight here that the cross-training of CandyCrunch on all glycan classes yielded performance synergy, as a model only trained on O-glycans performed worse for predicting O-glycans (Supplementary Table 8; topology: 72.3% accuracy, structure: 66.9% accuracy) than the model trained on all classes. We posit that this was due to the structure-based loss function we used for training, as well as shared information between spectra of different classes, stemming from shared glycan motifs across classes (for example, Neu5Ac-Hex).

The speed and relatively low resource requirements of CandyCrunch (Supplementary Fig. 13a) mean that samples can be

**Fig. 2 | Discovering diagnostic fragmentation using CandyCrumbs.**
**a**, Schematic view of the CandyCrumbs workflow for automatic fragment ion annotation. **b–e**, Negative ion mode spectra of reduced glycans with prediction confidence between 0.9 and 1.0 for Fucα1-2Galβ1-3GalNAc/Galβ1-4GlcNAcβ1-3Fuc (**b**), Neu5Acα2-3Galβ1-3GalNAc/Galβ1-3(Neu5Acα2-6)GalNAc (**c**), GlcNAcβ1-3(Neu5Acα2-6)GalNAc/GalNAcα1-3(Neu5Acα2-6)GalNAc (**d**) and GlcNAcβ1-3(Neu5Gcα2-6)GalNAc/GalNAcα1-3(Neu5Gcα2-6)GalNAc (**e**) were averaged and juxtaposed. Fragments exhibiting differential abundance were labeled by CandyCrumbs in the Domon–Costello nomenclature[30]. **f,g**, Negative

ion mode spectra of reduced glycans with prediction confidence between 0.6 and 1.0 for Neu5Acα2-3Galβ1-4GlcNAcβ1-2Manα1-3(Neu5Acα2-6Galβ1-4GlcNAcβ1-2Manα1-6)Manβ1-4GlcNAcβ1-4GlcNAc/Neu5Acα2-3Galβ1-4GlcNAcβ1-2Manα1-3(Neu5Acα2-6Galβ1-4GlcNAcβ1-2Manα1-6)Manβ1-4GlcNAcβ1-4GlcNAc (**f**) and Neu5Acα2-6Galβ1-4GlcNAcβ1-2Manα1-3(Galβ1-4GlcNAcβ1-2Manα1-6)Manβ1-4GlcNAcβ1-4GlcNAc/Neu5Acα2-3Galβ1-4GlcNAcβ1-2Manα1-3(Galβ1-4GlcNAcβ1-2Manα1-6)Manβ1-4GlcNAcβ1-4GlcNAc (**g**) were averaged, juxtaposed and labeled similar to **b–e**. Doubly charged fragment ions are colored gray.

exhaustively analyzed, without practical constraints to the most abundant structures, which is a routine necessity in human analysis. In its typical application, CandyCrunch also makes fewer assumptions about

what is or should be present in a sample, enhancing the chances for novel discoveries. This means, for example, that co-released *N*-glycans can be detected in *O*-glycan preparations (Supplementary Fig. 14).

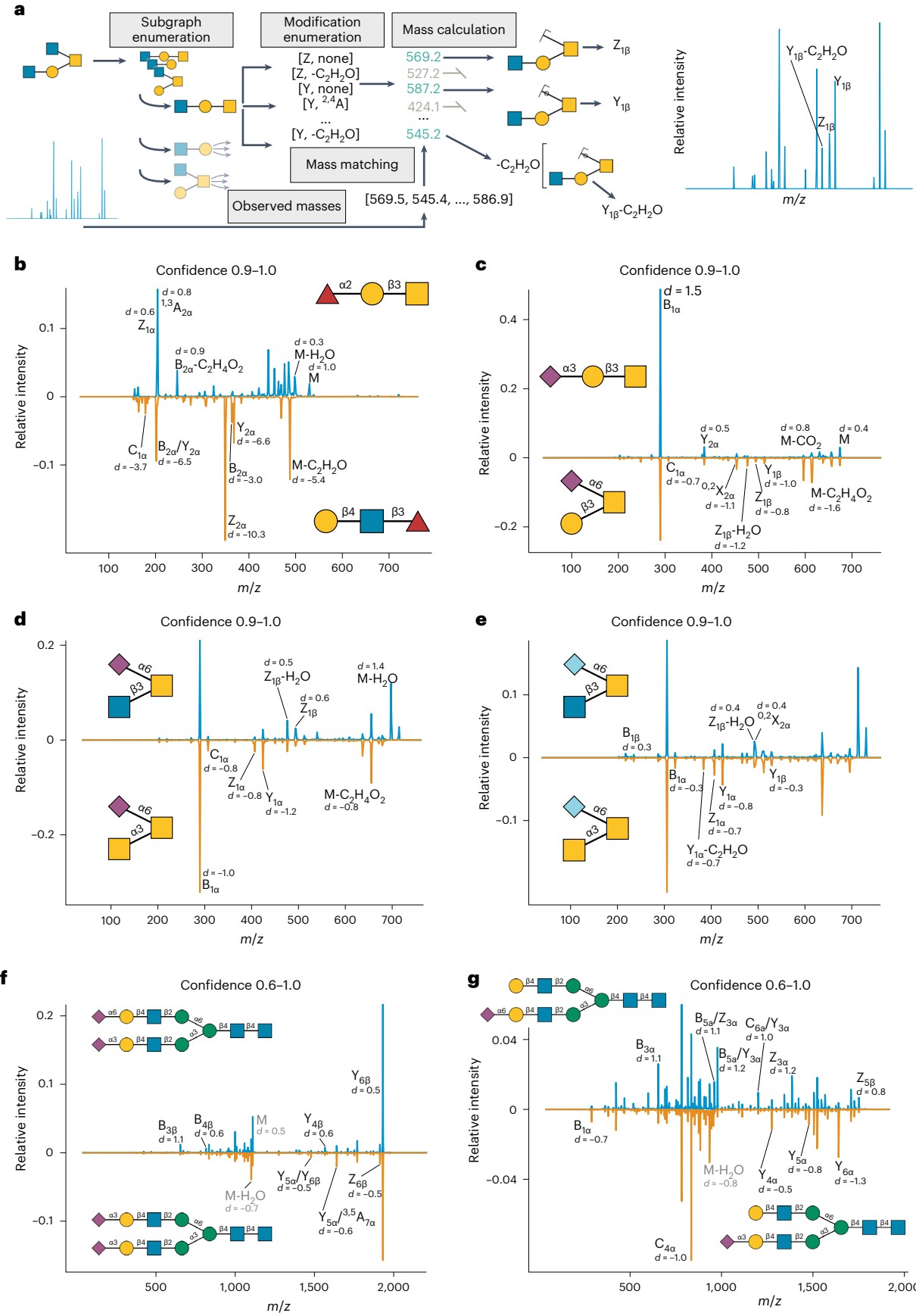

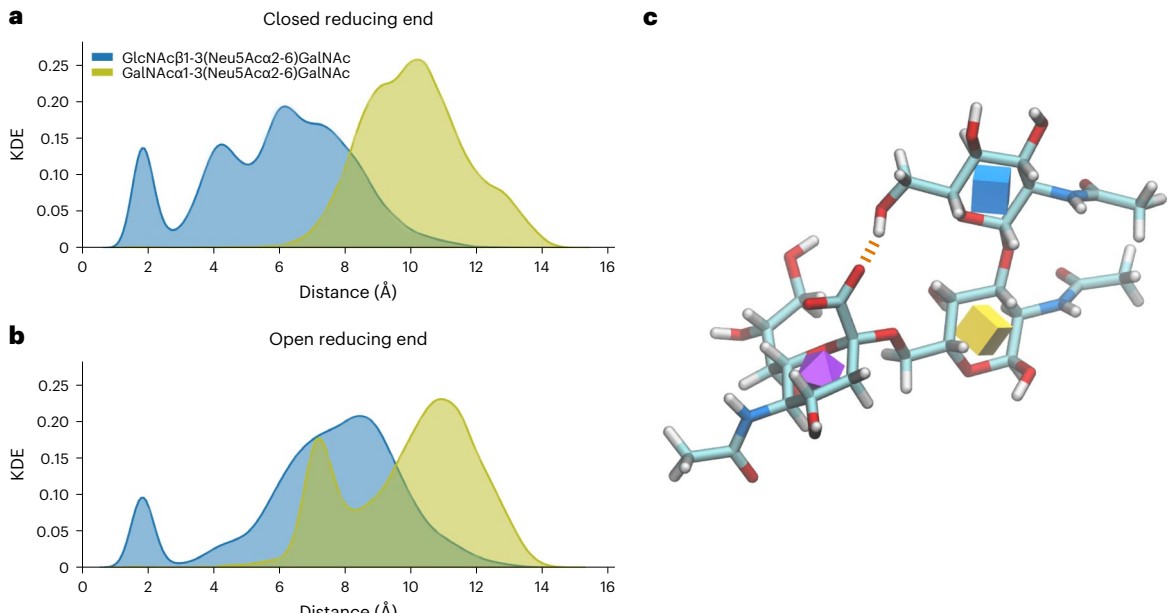

**Fig. 3 | Molecular dynamics reveals fragmentation mechanism. a,b**, Kernel density estimate distribution of the distance between the center of geometry of the carboxyl group of the sialic acid and the hydrogen of the hydroxyl group of C6 of the terminal HexNAc residues for the closed (**a**) and open (**b**) reducing GalNAc residue for both GlcNAcβ1-3(Neu5Acα2-6)GalNAc (blue) and GalNAcα1-3(Neu5Acα2-6)GalNAc (yellow green). The plots show how in

GlcNAcβ1-3(Neu5Acα2-6)GalNAc, the carboxyl group is able to interact with the hydroxyl of the C6 of the HexNAc. However, this interaction is not observed in GalNAcα1-3(Neu5Acα2-6)GalNAc. **c**, A representative snapshot of the structure of GlcNAcβ1-3(Neu5Acα2-6)GalNAc is shown, with the interaction between the two moieties displayed by a dashed line (orange). KDE, kernel density estimation.

## CandyCrumbs facilitates automated diagnostic ion discovery

When analyzed by humans, fragment ions are usually annotated via the Domon–Costello nomenclature[30] and used for elucidating the structure of a glycan. While there are programs that automate this assignment[21,22], they either are only accessible via graphical user interfaces or only provide annotations for simple fragment ions. We thus decided to implement an exhaustive Python-based solution to this problem, CandyCrumbs, which is also freely available via the CandyCrunch Python package. Given a candidate glycan sequence and fragment peaks, CandyCrumbs can automatically and rapidly (Supplementary Fig. 13b) annotate fragment ions in Domon–Costello and International Union of Pure and Applied Chemistry (IUPAC)-condensed nomenclature (Fig. 2a and Methods). Compared with alternative approaches, this presents the most feature-complete and rapid implementation of this task (Supplementary Fig. 10b).

Further, we used several domain knowledge-inspired heuristics and probability rules to highlight the most probable fragments (Supplementary Fig. 15 and Methods), if multiple fragmentation options could result in an *m/z* value that was acceptable at a given threshold. We then also integrated CandyCrumbs within the aforementioned open-access Colab notebook (at https://colab.research.google.com/github/BojarLab/CandyCrunch/blob/main/CandyCrunch.ipynb) for full flexibility. Our implementation of CandyCrumbs then allowed us to use it in a high-throughput setting and integrate it into CandyCrunch workflows, such as for identifying diagnostic ions at scale as discussed below, to aid expert annotation of challenging cases.

Reference spectra are routinely used as high-quality examples in semi-manual annotation[31]. As 'spectrum quality' is an ill-defined and subjective characteristic, we aimed to quantify this aspect by using calibrated[32] prediction confidence of CandyCrunch as a proxy, with the reasoning that a more confidently assessed spectrum is a higher-quality spectrum with more information for effective prediction. Rather than one reference spectrum, that is, the usual approach[31], we then extracted hundreds to thousands of high-quality spectra for a given structure from our dataset and engaged in highly powered statistical

comparisons between isomers. This identified numerous diagnostic ions and/or ratios for topologically distinct (Fig. 2b,c) and identical (Fig. 2d,e) isomers, with large effect sizes. This also extended to other glycan classes and, for example, facilitated detecting conserved fragmentation differences of linkages (for example, stronger B₃ ion in α2-6 versus α2-3) across glycan backbones (Fig. 2f,g) and recapitulated known effects from the literature[33], such as a higher stability of α2-6 versus α2-3 in negative mode (see B₁ ion in Fig. 2g). Importantly, these differences diminished, and eventually vanished, with lower-quality spectra (Supplementary Fig. 16). We then analyzed the predictiveness of these diagnostic features when reducing spectrum quality. Intriguingly, some diagnostic features, even if they were not the strongest initial signal, remained predictive even for medium- to low-quality spectra (Supplementary Fig. 17), making them promising candidates for aiding annotation.

Similarities between Neu5Ac and Neu5Gc versions of the same isomers (Fig. 2d,e) suggested molecular determinants of fragmentation propensities. We thus first analyzed all high-quality *O*-glycan spectra juxtaposing composition-matched glycans containing GalNAcα1-3 or GlcNAcβ1-3, confirming systematic fragmentation propensities on a global scale (Supplementary Fig. 18).

## Molecular dynamics supports diagnostic fragmentation

In the abovementioned scenario (Fig. 2d,e), our conclusion was that GlcNAcβ1-3(Siaα2-6)GalNAc fragmented along the HexNAc-HexNAc axis, while GalNAcα1-3(Siaα2-6)GalNAc fragmented along the Sia-HexNAc linkage. To elucidate how structural properties of these molecules could give rise to these differences in fragmentation behavior, we engaged in molecular dynamics simulations of both isomers.

The fragmentation pattern of the GlcNAcβ1-3(Siaα2-6)GalNAc glycan displayed evidence of a charge-induced fragmentation mechanism (Fig. 2d,e). In agreement with this, we saw evidence of the carboxylic acid moiety of the terminal sialic acid interacting with the hydrogen of the C6 hydroxyl group of the terminal HexNAc sugar

(Fig. 3). The interaction sampled 11.9% of our cumulative 2-μs simulations of GlcNAcβ1-3(Neu5Acα2-6)GalNAc. As these simulations were conducted in aqueous solution, rather than a vacuum as would be the environment for fractionation, the frequency of this interaction will be far greater during the in vacuo fragmentation due to absence of water molecules competing for hydrogen bonding. Therefore, this suggests that the charge-induced fragmentation mechanism of GlcNAcβ1-3(Neu5Acα2-6)GalNAc is due to removal of a proton from the terminal HexNAc sugar, therefore resulting in fragmentation along the HexNAc-HexNAc axis.

Conversely, simulations of GalNAcα1-3(Neu5Acα2-6)GalNAc were not able to sample this interaction (occurrence < 0.1%). As a result, fragmentation of this glycan occurs along the Neu5Ac-HexNAc linkage instead.

Furthermore, during the ionization of both of the glycans, reductive β-elimination would result in the reducing end GalNAc being reduced to an alditol. As this linearized structure may result in increased flexibility, we also conducted molecular dynamics simulations of both glycans with a linearized reducing GalNAc. These simulations yielded a similar insight to those described previously. In the reduced GlcNAcβ1-3 (Neu5Acα2-6)GalNAc glycan, the carboxyl group of the terminal sialic acid interacted with the hydrogen of the C6 hydroxyl group of the terminal HexNAc sugar during 6.8% of the simulated time. Again, the reduced GalNAcα1-3(Neu5Acα2-6)GalNAc was not able to sample this interaction (occurrence < 0.1%).

We therefore concluded that the identified fragmentation behavior can be used to distinguish between these two isomers, an endeavor that is otherwise challenging without specific enzymatic digestion. This implied that we could use our CandyCrunch and CandyCrumbs-powered approach to distinguish very close structural isomers based on diagnostic fragmentation behaviors, beyond single diagnostic ions or ratios and more akin to how human experts would distinguish them.

## New biological insights via CandyCrunch and CandyCrumbs

Striving towards AI-assisted glycomics, we propose our platform as a means to enhance human analysts by (1) saving time, (2) making annotations more robust and (3) analyzing samples more comprehensively. We illustrate the latter point with de novo predictions of murine intestinal glycans that were too low in abundance to be included in the original annotation but revealed, for example, the presence of Neu5Gc-containing glycans and low levels of sialyl-Tn antigen in these samples (Supplementary Fig. 19). Importantly, we do not claim that human analysts could not have annotated these structures in principle, but rather that very real time and resource constraints make this frequently infeasible in practice. This limitation is lifted by CandyCrunch.

To demonstrate that we could apply our developed methods to truly novel samples, we analyzed the serum *N*-glycome of

southern bluefin tuna (*Thunnus maccoyii*), which was measured within GPST000182 (ref. 34) but never reported in an annotated manner. This resulted in over 50 glycans, including high-mannose, hybrid and complex structures, with features such as bisecting GlcNAc, core and antenna fucosylation, Neu5Gc and multi-antennae *N*-glycans (Fig. 4a and Supplementary Fig. 20). In our comprehensive database within glycowork, not a single glycan from *T. maccoyii* has been reported so far, demonstrating that these pipelines can facilitate new discoveries.

We also wanted to highlight how predictions could be used downstream to derive new insights from aggregating glycomics studies. This can even be done in the context of already performed glycomics experiments, distilling results from the accumulated data of many years of study. For this, we re-used the total 250,000 *O*-glycan spectra mentioned in the context of Fig. 2 to construct biosynthetic networks[7]. As described above, this process filled in the gaps of unobserved intermediates in the biosynthesis of observed structures. A key benefit here is that all datasets have been analyzed by the same annotator (CandyCrunch), eliminating an important source of heterogeneity[35]. Applied to our dataset, this resulted in 1,003 biosynthetic networks (corresponding to 1,003 glycomics experiments measuring *O*-glycans) that we used to analyze systematic effects in that glycan class. This revealed that some intermediates were never measured (Supplementary Fig. 21 and Supplementary Table 9), such as the reducing end GalNAc (likely due to the mass range of the mass spectrometer used), while others, such as Gal3Sβ1-3GalNAc, were nearly always reliably measured whenever larger structures that included this building block as a substructure were present in a sample. We believe that this approach might shed light on subsets of the *O*-glycome that are currently hard to measure, as we here, again[7], noted the peculiar absence of GlcNAc-terminated structures from measured glycans as a trend.

Further analyses across our networks then allowed us to compare the reaction order of glycosyltransferases, reinforcing the highly dominant nature of galactosyltransferases[7] (Fig. 4b). Decomposing the biosynthetic networks into communities unveiled several conserved clusters that were modular and occurred in many of our datasets (Fig. 4c). Further investigation resulted in the observation that these clusters corresponded to the *O*-glycan core structures and their respective biosynthetic extensions (Fig. 4d). In general, these proved to be relatively modular, except for cases such as cores 1 and 2, which showed some biosynthetic overlap. We envision that this rapid decomposition of many networks into biosynthetic subcategories will prove useful for comparing and understanding the eventual terminal motifs that will be exposed in these different *O*-glycan cores, as well as their biosynthesis.

As a proof of concept, to demonstrate the capabilities of CandyCrunch for high-throughput analysis, we next predicted the *O*-glycomes of acute myeloid leukemia (AML) cell lines (GPST000214 (ref. 36)) and differentiated colorectal cancer cell lines (CaCo-2, GPST000256

**Fig. 4 | Deriving biological insights from CandyCrunch predictions. a**, Serum *N*-glycome of the southern bluefin tuna (*T. maccoyii*). Shown are the precursor ion intensities, arrayed by LC retention time. Representative structures that are meant to illustrate the identified sequence diversity are shown via the SNFG. Next to each structure, we show the cosine similarity of the shown spectrum and the averaged spectrum of all negative ion mode spectra of reduced glycans of the predicted structure with a confidence above 0.5 (see Fig. 2 for background). **b**, *O*-glycan reactions are path-dependent. For every situation in which two glycosyltransferases competed for the same substrate (*n* = 1,003 biosynthetic networks), we analyzed which order of reactions was experimentally observed across our networks. Box plots used the median as the center line and the 25th (Q1) and 75th (Q3) percentiles as the lower and upper edges of the box. The whiskers extend to the first data point within Q1 − 1.5 × IQR (interquartile range) and to the last data point within Q3 + 1.5 × IQR and outlier values outside this range are depicted as diamonds. **c**, *O*-glycan networks decomposed into biosynthetic communities relating to core structures. We detected communities

via the Louvain method and calculated their pairwise Jaccard distances, shown here as a hierarchically clustered heatmap. **d**, Community corresponding to core 5 *O*-glycans. Clustering of the distance matrix from **c** using OPTICS (Ordering Points To Identify the Clustering Structure)[46] resulted in conserved communities broadly corresponding to *O*-glycan cores, with the one from core 5 being shown here as a network, nodes scaled by degree. **e**, Clustering cancer cell line *O*-glycomes. Predicted *O*-glycomes of AML cell lines (GPST000214) and differentiated colorectal cancer cell lines (CaCo-2, GPST000256), via a CandyCrunch model not trained on these datasets, are shown via *t*-SNE (*n* = 103), using glycan abundance as features. **f**, Differential glycan expression between AML and colorectal cancer cell lines. Given the predicted glycomes of **e**, we used the get_volcano function from glycowork to test differential expression at the motif level (two-tailed Welch's *t*-test), shown as a volcano plot. Differentially expressed glycans are drawn inversely scaled by corrected *P* value (Holm–Šídák correction for multiple testing). FC, fold change.

(ref. 37)). With a total of 103 glycomics raw files for this analysis, we could show that the predicted glycomes of AML and colorectal cancer cell lines formed distinct clusters (Fig. 4e), which both were separate from the blanks used in GPST000256. We then engaged in a differential glycan expression analysis to investigate what distinguished these

clusters. While there was considerable intra-cluster heterogeneity, this analysis revealed that the colorectal cell lines on average were more enriched in structures containing fucosylated galactose and remnant *N*-glycans, while the AML cell lines exhibited higher levels of sialylated glycans and Lewis structures (Fig. 4f). This set of analyses shows that

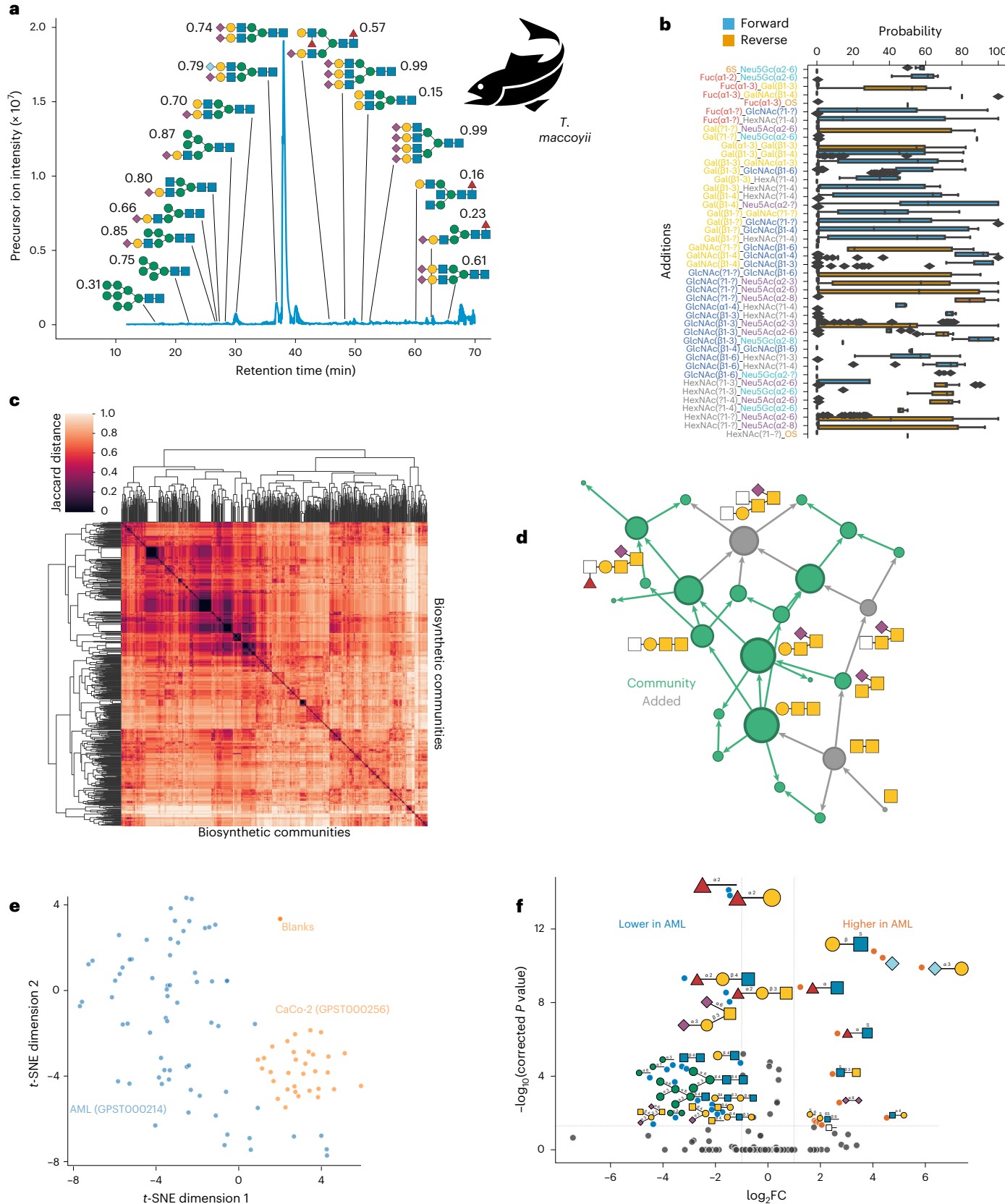

CandyCrunch can be applied to large sets of glycomics measurements and eventually be used in conjunction with other glycowork functionality to reveal dysregulated glycans and glycan motifs, directly from LC–MS/MS raw files.

## Discussion

We present here generalizable methods to (1) predict glycan structures from LC–MS/MS data using deep learning (CandyCrunch) and (2) automatically annotate fragment ions in higher-order tandem mass spectrometry spectra (CandyCrumbs). Proven performance on blinded data with ground truth labels (Fig. 1h, Supplementary Figs. 11, 12, 14 and 19 and Supplementary Table 8) cements the usefulness of CandyCrunch. Both CandyCrunch and CandyCrumbs are suited for high-throughput usage and can scale to large datasets as well as extremely diverse glycans and experimental setups. With the high performance that we demonstrate here, we are confident that these pipelines will be useful both for experts, accelerating and augmenting their workflows, as well as for less experienced users, similar to how automated workflows in other systems biology disciplines have democratized access to state-of-the-art methods[38,39]. We have demonstrable experience with maintaining software over longer periods (via our glycowork platform) and, since we ourselves are active users of CandyCrunch for our core research, have a natural incentive to further develop this technology.

Our approach is ultimately limited by the representativeness of available data. While CandyCrunch is applicable to all major glycan classes and most experimental setups (for now limited to electrospray ionization-type setups), we do note that the very best results can be expected for reduced glycans in negative mode, particularly *O*-glycans or free oligosaccharides. This is both a result of high-quality data in those cases and particular efforts in fine-tuning our pipeline for optimal results, as they intersected most with our own research interests and capabilities. In general, compelling results can be expected for samples similar to our training data, strongly enriched in mammalian and fish samples (Supplementary Fig. 22), and we expect to perform worse, on average, on remote samples such as from invertebrates. We envision that, with increasing data, this will improve. We thus urge the community to make their glycomics data (as well as high-quality annotations) available through platforms such as GlycoPOST[40], as this will improve approaches such as CandyCrunch, and ultimately advance glycobiology and its applications.

We recognize that, as with any model, CandyCrunch predictions are imperfect, exhibiting false negative and false positive predictions, which occasionally might not resemble errors made by humans. Particularly, non-CandyCrunch glycan additions within our pipeline, via biosynthetic networks and database queries, exhibit a more tentative character and should be further evaluated by experts. For ideal results, we always recommend predictions to be further refined by experts. We are, however, convinced that CandyCrunch predictions can raise result quality and comprehensiveness for both experts and novices, in addition to the considerable increase in throughput. Lastly, during data curation, we assumed expert annotations within our training data to be correct, which may retain analyst bias, such as preferential annotation of type II versus type I LacNAc structures in *N*-glycans without conclusive evidence. We do note, however, that the annotations that we trained on were, in part, informed by other sources of information, such as third-generation product ion spectra or exoglycosidase digestions. Once sufficient data become available, future work may also extend this approach to higher-order tandem mass spectrometry spectra and/or exoglycosidase treatments, with more detailed structural information.

Beyond the fact that the zero-shot capabilities of CandyCrunch are limited, we would also like to note that, while we support common derivatizations such as permethylation, we do not currently support every type of glycan modification within CandyCrunch and CandyCrumbs. Specialized methods, such as azidosugars[41], are at the moment beyond our scope. Once sufficient raw data of new modifications become available, CandyCrunch can be easily retrained (the CandyCrunch package includes a training script, and models can be retrained in less than 12 h on a free Google Colab instance).

We are also enthusiastic about the potential of upcoming methods to simulate high-resolution fragmentation spectra via deep learning[42], which could be adapted for AI-glycomics in future work and aid either training or the evaluation of prediction results. Further, once sufficient data from either high-resolution mass spectrometers or absolute normalizations of retention time (for example, via glucose units) become available, we expect CandyCrunch to reach even higher performance. While we focus on glycomics here, we envision that analogous efforts in glycoproteomics could also advance and accelerate the field. Overall, we conclude that our presented methods not only pave the way for AI-enhanced structural glycomics but also enable many other avenues ranging from systematic comparisons over data science to glycoinformatics. This is facilitated by our large, curated dataset and the ability to quantify spectrum quality, engaging in analyses at scale for many different aspects of glycomics data.

## Online content

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

## Methods

### Dataset

Tandem mass spectra from electrospray ionization experiments stemmed from repositories such as GlycoPOST[40], MassIVE, UniCarb-DB[31], UniCarb-DR and NIST, as well as from individual publications with associated public raw data. A full list of the 196 data sources can be found in Supplementary Table 10. All raw files were converted into the open-access format .mzML using the msconvert software[47]. A custom script using the pymzML package[48] (v.2.5.2) or pyteomics[49] (v.4.6) was used to extract all spectra at the MS/MS level, together with their stored precursor ion $m/z$ and retention time, if available. This extraction functionality is now available as the process_mzML_stack function within our CandyCrunch package (v.0.3.0), next to an analogous process_mzXML_stack function. We extracted up to 1,000 fragment peaks of the highest intensity per spectrum, if available. Then, spectra were retained that fell within ±0.5-Da $m/z$ and ±2-min retention time of reported glycan peaks in the associated publications. All retained spectra were kept for self-supervised training, paired with the information of the respective glycan class, while only spectra that could be unambiguously linked to structures described in the respective publications were kept for supervised training. This resulted in a total number of 625,547 glycan spectra, of which 489,103 spectra were labeled with a defined structure and could be used for training, the latter stemming from 3,391 unique glycan structures (Supplementary Table 11). The full dataset can be found at Zenodo under https://doi.org/10.5281/zenodo.10029271 (ref. 43).

### Data processing

We first removed all spectra with a retention time below 2 min as noise. Retention times then were normalized for each individual sample, by dividing absolute retention times by the respective maximal retention time (or a minimum of 30, if the maximum extracted retention time was below 30). Missing retention times were assigned a value of zero. Fragment intensities were normalized for each spectrum, by dividing the intensity of each peak by the total intensity of the spectrum. Then, intensities were binned in 2,048 equal-sized $m/z$ windows from the observed minimum (39.714) up to a maximum of 3,000. Additionally, the $m/z$ remainder (that is, the difference of the $m/z$ of the highest intensity peak of a bin to the left bin window) was calculated for each bin, as suggested in Altenburg et al.[25], allowing the model to learn exact peak location despite binning. We explicitly emphasize here that this procedure, combined with the fact that most bins contain only one peak (Supplementary Fig. 1), allows us to override the nominal mass resolution of 1.45 Da that our binning creates. Glycan class, MS ion mode, ion trap type, LC type and glycan modification type were coded as integers to allow for learned embeddings.

During training, we capped all glycan structures to at most 1,000 randomly sampled spectra per structure in the independent test set, to avoid imbalance in assessment by frequently observed but simple glycans. We used an 85/15 split into train/test set for the 489,103 spectra, which were split on the level of samples, to ensure that spectra of one sample were not found in both train and test sets and thus make the generalizability estimation more robust. For training, classes in the test set that would constitute zero-shot prediction were afterwards moved into the train set.

### Model architecture

CandyCrunch is a dilated residual neural network, with additional embedded inputs, to predict glycan structure from tandem mass spectrum in a multiclass classification setup.

For the processing of binned intensities and $m/z$ remainders, a one-dimensional convolution layer was followed by a leaky rectified linear unit (ReLU) and six residual dilated convolutions, with dilations of 1, 2, 4, 8, 16 and 32. Then, we used max-pooling with a kernel size of 20 and a fully connected layer to bring this output to a dimensionality of 1,024. Glycan class, MS ion mode, ion trap type, LC type and glycan modification type were embedded into dimensionalities of 24, respectively. Precursor $m/z$ and normalized retention time were also brought to dimensionalities of 24 via a fully connected layer, a layer normalization and a leaky ReLU. Then, all inputs were concatenated and passed through two sets of fully connected layers, layer normalization, leaky ReLUs and dropout (at a rate of 0.2). Finally, a last fully connected layer yielded the class probabilities. In total, CandyCrunch exhibited 12,375,084 trainable parameters.

### Model training

All models were trained in PyTorch[50] (v.2.1.0) using two Zotac GeForce RTX 4090 Trinity GPUs. CandyCrunch was initialized via He initialization. All models were trained for 200 epochs, with an early stopping regularization of stopping training after 12 epochs without improvement in the test loss and a batch size of 256.

We set the learning rate at 0.0001, with a schedule to reduce the learning rate to a fifth after four epochs with no improvement in test loss. As a base optimizer we used AdamW with a weight decay of $2 \times 10^{-5}$, which was further modified via adaptive sharpness-aware minimization (ASAM)[51] to ensure a generalizable final model.

Data augmentation during training was used only on the training set and included random (1) low-intensity peak removal, (2) peak intensity jitter and (3) new peak addition for the binned spectrum, as proposed previously for MS[52], as well as adduct formation of the precursor ion (acetate/sodium adducts) and random noise of the precursor $m/z$ (±0.5 Da) and retention time (±10%).

As our base loss, we used PolyLoss[53], with an additional label-smoothing of 0.1 and epsilon = 1. We note that the label-smoothing employed here, as well as the fact that the annotators for many of our datasets have used additional information to refine their annotations (for example, third-generation product ion spectra, exoglycosidases), at least in part counteracts potential concerns about label uncertainty. We also used two additional loss terms, informed by domain knowledge, that were added to the PolyLoss term. These constituted a structure distance loss and a composition distance loss. Both involved the calculation of a distance matrix, based on pairwise cosine distances of fingerprint vectors of either the number of mono- and disaccharide motifs or the base composition of two glycans. All operations on glycans were performed using glycowork[29] (v.1.1.0). Then, the class probabilities for each input sample, transformed via a softmax activation, were multiplied by the structure distance vector and the composition distance vector (that is, the distance to the target glycan), followed by mean averaging to obtain loss terms. This unsupervised procedure preferentially penalizes confidently predicted but structurally dissimilar glycans and improves performance as well as the meaningfulness of errors.

We first engaged in supervised training on annotated MS/MS spectra. Then, using the trained model we predicted glycan structure for our unannotated spectra for self-supervised training. Spectra with a prediction score of over 0.7 were then merged with the original training dataset, followed by a deduplication step. Specifically, as described above, we retained the same test set and again formed a training dataset with at most 1,000 examples per glycan in the independent test set, followed by re-training.

### Model inference

To predict glycan structures from unannotated raw files, all tandem spectra were extracted via pymzML as described above and processed as described for the general data processing. Then, we grouped $m/z$ precursor ions by scanning for discontinuities larger than 0.5 Da in the extracted spectra. Within these $m/z$ groups, we searched for structural isomers by analyzing their retention time in chunks of 0.5 min. While this may lead to overlaps between isomer peaks, this is not an inherent problem, as long as co-elution is not perfect, as different chunks will

still retain the respective isomers as the dominant species, which will be reflected in the final output table. For each retention time group, we averaged all spectra for input of a robust averaged spectrum to CandyCrunch and extracted the median spectrum, to have a representative spectrum for each glycan entity in the sample. We first retrieved the top 25 predictions for each averaged spectrum, using the trained CandyCrunch model. We then employed a single-parameter variant of Platt Scaling[32] to calibrate the prediction confidence before the softmax layer, using a scaling factor of 1.15 that was estimated via the limited-memory Broyden–Fletcher–Goldfarb–Shanno algorithm. Using test-time augmentation, we averaged the predictions of five independent inferences that were modified with the same data augmentation strategy as employed during training.

Next, we used domain knowledge to automatically filter out predictions, such as of (1) a prediction probability below a threshold of 0.01, (2) the wrong glycan class, (3) the wrong mass, even when considering multiply charged ion forms, and (4) predictions that lacked corroborating diagnostic ions in their fragment lists. Domain-specific exceptions were made, such as allowing cross-class predictions if the prediction confidence was extraordinarily high (above 0.2; justified by the fact that *O*-linked glycan samples often contain remnant *N*-linked glycans and so on) Finally, predictions were deduplicated by merging any mass/retention windows that resulted in identical predictions.

Lastly, we used biosynthetic knowledge to refine our predictions, conceptualized in the canonicalize_biosynthesis function within CandyCrunch. Using the subgraph_isomorphism function from glycowork and starting from the largest glycan prediction, we searched for top-1 predictions of biosynthetic precursors in the whole prediction dataframe. For each prediction at mass $M$, we added 0.1 to its prediction confidence for each unique biosynthetic precursor in top-1 predictions at mass $M$-1, $M$-2, …, $M$-$n$. If this changed the order of predictions, we re-ordered predictions according to their scores. Thereafter, scores were re-normalized to 1 and the, up to, top-5 predictions were retained. This procedure not only improved the accuracy of our results but also increased the meaningfulness and consistency of both correct and wrong predictions (that is, wrong predictions were structurally closer to the ground truth after this procedure).

Spectra without valid predictions but with valid compositions, cross-referenced by relevant databases within glycowork, were also retained and subjected to as many of the abovementioned domain filters as possible. Whenever available, top-1 predictions were paired with their GlyTouCan ID[54]. The whole inference workflow, including elements described below, is available via the wrap_inference function in the CandyCrunch package. Available options for running the function are shown in Supplementary Table 12 and mentioned in the documentation of the CandyCrunch package (https://github.com/BojarLab/CandyCrunch).

For the case of multiple samples from the same experiment, we also added the wrap_inference_batch function to the CandyCrunch package. This expanded workflow aligns retention times across samples, if possible and suitable, to build a prediction library and ensure that shifts in retention time between samples are accommodated.

## Zero-shot prediction
For a given sample, all retained top-1 predictions were used to construct a biosynthetic network as described previously[7], using the implementation within glycowork. For milk oligosaccharides, this also included evolutionary pruning, as pre-calculated species networks were available. Then, we calculated whether any of the inferred biosynthetic precursors would explain the mass and composition of glycan spectra without a valid prediction. Matches within a mass difference of 0.5 Da, including multiply charged ions, were retained as additional predictions beyond our model-defined library of predictable glycans. While direct model predictions were awarded the evidence

category 'strong', the biosynthetic network intermediaries merited the category 'medium'.

Next, we checked for missed Neu5Gc-substituted Neu5Ac-glycans and vice versa (that is, a mass difference of 16 Da per substitution, with the corresponding diagnostic ions). Similarly, in the case of an *O*-glycan sample, we checked for missed GlcNAc6S-substituted GlcNAc-glycans and vice versa (connected to the reducing end GalNAc). Additionally, we used a suitable subset of the glycowork-stored database, of the right taxonomic section and glycan class, to search for possible matches to compositions without predictions. Both of these endeavors were annotated with the evidence label 'weak'.

After these additional routines to enable predictions outside of our defined list of glycans, we again employed the domain knowledge-informed filters mentioned above. This ensured that glycans introduced via these methods still had empirical support in the underlying data. Predictions from these routines were also subjected to the canonicalize_biosynthesis workflow from above (although 'bonus' points were awarded only for biosynthetic precursors from the 'strong' category), to allow for prioritization of the most probable structures.

## Fragment annotation via CandyCrumbs
The final prediction of the CandyCrunch model was used as a starting point for fragment annotation and converted into a directed graph using NetworkX (v.3.0), each monosaccharide making up a node and each linkage labeling an edge. The randomized enumeration method was implemented to find all induced connected subgraphs[55]. After filtering which modifications were physically possible based on linkage numbers, each terminal monosaccharide on the subgraphs was permuted to create these cross-ring or bond fragmentations. Each possible global modification was also added to each fragment. The mass of each theoretical fragment was calculated to then be matched with observed masses in MS/MS spectra. Finally, the fragments were converted into Domon–Costello[30] and IUPAC-condensed nomenclature. If multiple fragment possibilities could explain a given $m/z$ value, a prioritization scheme was developed (Supplementary Fig. 15), which emphasized prior likelihood of each fragment option and the evidence of the remaining fragments in a given tandem spectrum. We note that fragment prioritization is an optional step in this workflow and can be disabled, if all possible fragments are desired. CandyCrumbs is available via CandyCrunch.analysis.CandyCrumbs in our developed Python package.

## Molecular dynamics simulation
Initial conformations for the GlcNAcβ1-3(Neu5Acα2-6)GalNAc and GalNAcα1-3(Neu5Acα2-6)GalNAc glycans were obtained using the Carbohydrate Builder tool of the GLYCAM-Web server[56]. Four structures were produced for each glycan with different combinations of the α2-6 torsion angles. This approach provided different initial starting points for the simulations, and thus maximized the sampling of the conformational space. Each glycan was parameterized with the GLYCAM06-j1 forcefield[57], and a cuboid solvent box of TIP3P water molecules created to produce a minimum solute distance of 15 Å. In the case of the reduced glycan structures, the structures of the open GalNAc were parameterized using the GAFF2 forcefield[58]. A single Na$^+$ ion was included in each system to neutralize the net charge of the system. These systems were then converted into GROMACS topology files using Acpype[59]. For each initial starting conformation of each system, a 500-ns simulation was performed using GROMACS2022.4 (ref. [60]), resulting in 2 μs of simulations for each respective system.

## Biosynthetic network analysis
For all networks constructed and analyzed in this work, we used the code functionality within the glycowork.network.biosynthesis module (v.1.1.0). Our analyses were oriented very closely by the ones described by Thomès et al.[7] Briefly, the analysis of glycosyltransferase

competition was performed by analyzing diamond-like network motifs via the trace_diamonds and find_diamonds functions within glycowork. Thereby, we analyzed the proportion of networks that presented a certain case of glycosyltransferase competition and counted how often each alternative order of reactions was experimentally observed among these. This allowed us to analyze which reaction order dominated across (1) glycan contexts and (2) networks. The differences shown in Fig. 4 were further filtered to contain at least (1) two glycan sequence contexts, (2) a mean difference of 30 and (3) a corrected *P* value below 0.01.

Biosynthetic communities were extracted using the get_communities function, from glycowork, on reaction path preference-pruned biosynthetic networks[7]. Conserved communities were detected by first calculating a distance matrix based on pairwise Jaccard distances, followed by clustering these distances using the OPTICS algorithm as implemented in scikit-learn (v.1.2.2), with a minimum number of 50 samples per cluster.

### Statistical analyses
Comparing two groups was done via one-tailed or two-tailed Welch's *t*-tests. In all cases, significance was defined as $P < 0.05$. All multiple testing was corrected with a Holm–Šídák correction. All statistical testing has been done in Python 3.9 using the statsmodels package (v.0.13.5) and the scipy package (v.1.10.1). Effect sizes were calculated as Cohen's *d* using glycowork (v.1.1.0). The correlation of distance matrices was performed via two-sided Mantel tests as implemented within scikit-bio (v.0.5.8).

### Reporting summary
Further information on research design is available in the Nature Portfolio Reporting Summary linked to this article.

### Data availability
All relevant data, including their data provenance with accession IDs from GlycoPOST[40], MassIVE, UniCarb-DB[31], UniCarb-DR or NIST, can be found at Zenodo via https://doi.org/10.5281/zenodo.10029271 (ref. 43) or are contained within Supplementary Tables 10 and 11. The 196 data sources are listed in Supplementary Table 10.

### Code availability
All relevant code is integrated into glycowork (v.1.1.0) and/or can be found at https://github.com/BojarLab/CandyCrunch. CandyCrunch and CandyCrumbs can also be readily accessed at https://colab.research.google.com/github/BojarLab/CandyCrunch/blob/main/CandyCrunch.ipynb.

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

### Acknowledgements
This work was funded by a Branco Weiss Fellowship–Society in Science awarded to D.B., by the Knut and Alice Wallenberg Foundation and by the University of Gothenburg, Sweden. The Science Foundation of Ireland (SFI) Frontiers for the Future Programme is gratefully acknowledged for financial support of C.M.I. (grant no. 20/FFP-P/8809). We also thank C. Fogarty for his assistance in parameterizing the open GalNAc residue for molecular dynamics simulations. We thank SciLifeLab and BioMS (Swedish research council) for providing financial support to the Proteomics Core Facility, Sahlgrenska Academy. The funders had no role in study design, data collection and analysis, decision to publish or preparation of the manuscript.

### Author contributions
D.B. and J.U. conceived the method. D.B. curated the dataset. D.B., E.F., C.M.I. and J.U. performed computational analyses. D.B., C.M.I. and J.U. prepared the figures. C.J., N.G.K. and K.A.T. confirmed spectra annotations and provided domain expertise for method development and application. D.B. and E.F. supervised. All authors wrote and edited the paper.

### Funding

### Competing interests
The authors declare no competing interests.

### Additional information

**Correspondence and requests for materials** should be addressed to Daniel Bojar.

# Reporting Summary

## Statistics

For all statistical analyses, confirm that the following items are present in the figure legend, table legend, main text, or Methods section.

| n/a | Confirmed | |
|---|---|---|
| ☐ | ☒ | The exact sample size ($n$) for each experimental group/condition, given as a discrete number and unit of measurement |
| ☒ | ☐ | A statement on whether measurements were taken from distinct samples or whether the same sample was measured repeatedly |
| ☐ | ☒ | The statistical test(s) used AND whether they are one- or two-sided *Only common tests should be described solely by name; describe more complex techniques in the Methods section.* |
| ☒ | ☐ | A description of all covariates tested |
| ☐ | ☒ | A description of any assumptions or corrections, such as tests of normality and adjustment for multiple comparisons |
| ☐ | ☒ | A full description of the statistical parameters including central tendency (e.g. means) or other basic estimates (e.g. regression coefficient) AND variation (e.g. standard deviation) or associated estimates of uncertainty (e.g. confidence intervals) |
| ☐ | ☒ | For null hypothesis testing, the test statistic (e.g. $F$, $t$, $r$) with confidence intervals, effect sizes, degrees of freedom and $P$ value noted *Give P values as exact values whenever suitable.* |
| ☒ | ☐ | For Bayesian analysis, information on the choice of priors and Markov chain Monte Carlo settings |
| ☒ | ☐ | For hierarchical and complex designs, identification of the appropriate level for tests and full reporting of outcomes |
| ☐ | ☒ | Estimates of effect sizes (e.g. Cohen's $d$, Pearson's $r$), indicating how they were calculated |

*Our web collection on statistics for biologists contains articles on many of the points above.*

## Software and code

Policy information about availability of computer code

| Data collection | Python (version 3.9), glycowork (version 1.1.0), CandyCrunch (version 0.3.0), pymzML (version 2.5.2), pyteomics (version 4.6), msconvert (version 3.0.23158-5fbe5ea) |
|---|---|
| Data analysis | Python (verson 3.9), glycowork (version 1.1.0), CandyCrunch (version 0.3.0), PyTorch (version 2.1.0), NetworkX (version 3.0), scikit-learn (version 1.2.2), statsmodels (version 0.13.5), scipy (version 1.10.1), scikit-bio (version 0.5.8), https://colab.research.google.com/github/BojarLab/CandyCrunch/blob/main/CandyCrunch.ipynb, https://github.com/BojarLab/CandyCrunch |

For manuscripts utilizing custom algorithms or software that are central to the research but not yet described in published literature, software must be made available to editors and reviewers. We strongly encourage code deposition in a community repository (e.g. GitHub). See the Nature Portfolio guidelines for submitting code & software for further information.

## Data

Policy information about availability of data

All manuscripts must include a data availability statement. This statement should provide the following information, where applicable:

- Accession codes, unique identifiers, or web links for publicly available datasets
- A description of any restrictions on data availability
- For clinical datasets or third party data, please ensure that the statement adheres to our policy

All relevant data, including their data provenance with accession IDs from GlycoPOST, MassIVE, UniCarb-DB, UniCarb-DR, or NIST, can be found at Zenodo under the doi:10.5281/zenodo.10029271 or is contained within the supplementary material. The 196 data sources are listed in Supplementary Table 10.

## Human research participants

Policy information about studies involving human research participants and Sex and Gender in Research.

| | |
|---|---|
| Reporting on sex and gender | n/a |
| Population characteristics | n/a |
| Recruitment | n/a |
| Ethics oversight | n/a |

Note that full information on the approval of the study protocol must also be provided in the manuscript.

# Field-specific reporting

Please select the one below that is the best fit for your research. If you are not sure, read the appropriate sections before making your selection.

☒ Life sciences    ☐ Behavioural & social sciences    ☐ Ecological, evolutionary & environmental sciences

For a reference copy of the document with all sections, see nature.com/documents/nr-reporting-summary-flat.pdf

# Life sciences study design

All studies must disclose on these points even when the disclosure is negative.

| | |
|---|---|
| Sample size | All publicly available LC-MS/MS glycomics data using electrospray ionization were gathered from all available sources (cut-off September 2023). This resulted in 625,547 glycan spectra. |
| Data exclusions | We excluded spectra which we could not unambiguously pair with an expert glycan annotation. This resulted in 489,103 spectra that were labeled with a defined structure. |
| Replication | All results could be successfully replicated for at least three times, for instance with the same model trained with different seeds. |
| Randomization | Training and validation data were split randomly on a file basis, so that no spectra of the same raw file could be found in both training and validation set, improving robustness. |
| Blinding | For all assessments of model performance, experimenters were blinded to the expert annotation, which was only revealed at the time of evaluation. |

# Reporting for specific materials, systems and methods

We require information from authors about some types of materials, experimental systems and methods used in many studies. Here, indicate whether each material, system or method listed is relevant to your study. If you are not sure if a list item applies to your research, read the appropriate section before selecting a response.

## Materials & experimental systems

| n/a | Involved in the study |
|-----|----------------------|
| ☒ | Antibodies |
| ☒ | Eukaryotic cell lines |
| ☒ | Palaeontology and archaeology |
| ☒ | Animals and other organisms |
| ☒ | Clinical data |
| ☒ | Dual use research of concern |

## Methods

| n/a | Involved in the study |
|-----|----------------------|
| ☒ | ChIP-seq |
| ☒ | Flow cytometry |
| ☒ | MRI-based neuroimaging |

