## [Peer Review File · Nature Methods]

Peer Review Information

Manuscript Title: Predicting glycan structure from tandem mass spectrometry via deep learning

Corresponding author name(s): Daniel Bojar

Editorial Notes: None

Reviewer Comments & Decisions:

Decision Letter, initial version:

Dear Dr. Bojar,

Your Article entitled "Predicting glycan structure from tandem mass spectrometry via deep learning" has now been seen by 2 reviewers, whose comments are attached. In the light of their advice we have decided that we cannot offer to publish your manuscript in Nature Methods.

You will see that, while they find your work of some potential interest, the reviewers raise concerns about the advance your methodological approach represents over available methods and about its broad applicability at this stage. Both reviewers raise the issue of accuracy and generalizability of the approach as well as concerns about potentially severe limitations in broader applicability. We think that these criticisms are sufficiently important as to prevent publication of your work in Nature Methods.

Although we cannot offer to publish your manuscript, I suggest that you consider Nature Communications as a suitable venue for this work. To transfer your manuscript, please use our manuscript transfer portal. You will not have to re-supply manuscript metadata and files, unless you wish to make modifications. For more information, please see our manuscript transfer FAQ page.

I am sorry that we cannot be more positive on this occasion but hope that you find the reviewers' comments helpful when preparing your paper for submission elsewhere.

Sincerely,
Arunima

Arunima Singh, Ph.D.
Senior Editor
Nature Methods

Reviewer Comments:

Reviewer #1 (Remarks to the Author):

There are serious concerns about this manuscript. It is well established that existing tandem MS methods do not suffice to assign complete glycan topologies. That glycan epitopes have diagnostic intensity ratios does not equate to the ability to assign the complete glycan structure. The mass spectral data do not contain sufficient information to define the glycan structure. It appears that the authors focus on differentiating glycan isomers using their algorithm. The challenge here is that different tandem MS modes produce different diagnostic ions. Considering data produced using a single tandem MS mode, there are considerable variations in dissociation conditions among data sets that make identification of consistent product ion ratios very difficult. In the protein folding field, AlphaFold succeeded because of the large set of high-quality protein three dimensional structures available to train the algorithm. By contrast, the glycomics field has yet to achieve consensus regarding how complete glycan structure assignments can be made from tandem mass spectra. The available data do not constitute high quality assignments necessary to train an algorithm. It is therefore not clear whether the CandyCrunch predictions are meaningful since they may reflect biases inherent in the raw data rather than structural determinants. To resolve these concerns, it is necessary for the authors to analyze new data on known glycans to illustrate the performance of the algorithm. These experiments must be performed blinded. The examples showing that complex N-glycome data can be annotated do not suffice since there is no ground truth. The authors must use ground truth samples to demonstrate the effectiveness of their algorithm.

Reviewer #3 (Remarks to the Author):

Urban et al. present a deep learning model for the automated prediction of glycan structures from LC-MS/MS data. Such tools are seriously needed in the field, and are especially valuable when easy to use, providing accurate information about prediction certainty, and insight in the spectra annotation to allow expert evaluation. The tool presented here is highly promising in addressing these points. After reading the manuscript (clearly written, although the authors might have another look to simplify the text by using less complex wording) and trying the tool via the google drive-based interface, overall evaluation of it is positive, however the general feeling arises that the publication of the tool is slightly rushed and not all functionality is as good as envisioned/described in the text, also some unclarities and points for improvement remain:

1) The tool was evaluated by the reviewer on PGC-MS data (negative mode) of reduced N-, O- and GSL-glycans, as this was the main training data described in the manuscript. The tool was considered relatively easy to use, although multiple file name recognition errors were encountered which appeared to be resolved by refreshing the page multiple times. Also, while indicated that mzXML would be a suitable format, in our hands only mzML files could be run. While the tool was able to pick up large part of the glycans that were also manually annotated, some of them remained unidentified, especially observed for a couple of isomeric structures. On the other hand, some new, low abundant structures were found that were not picked up manually. Finally, some structures were reported that could not be found back in the data when evaluated manually. All in all this shows that the tool is valuable in the data analysis process and can be used as a first step in the data analysis workflow, however it remains necessary to manually/by another tool do a MS1 peak picking to see what potential glycan signals are not annotated (see comment 6) and to manually evaluate, especially the

low scoring glycans.

2) To enable this manual evaluation a couple of things are essential in the output of the tool, including the compositional annotation of the glycan often used in MS data, indicating the number of hexoses, hexnacs, fucoses etc in the glycan (eg. H4N5F1 or Hex4HexNAc5Fuc1), the theoretical mass of the annotated glycan and the ppm error between the data and the structure found. Importantly, it should be easy to look at the fragmentation spectrum used, including the annotated peaks for the glycan fragments (see comment 3).

3) The current solution to evaluate the annotated fragmentation spectrum (CandyCrumbs) is not sufficient to allow an easy look at the data. Mainly because:

- a. Multiple peaks are annotated with the same fragments.
- b. You only see the fragments when you mouse-over a peak and often the content of the mouse-over is not readable because of some screen dimension issues.
- c. There is no indication of the ppm error between the theoretical fragment and the observed peak, making it very difficult to evaluate the trueness of an annotation. It would be good if the mass accuracy of the used method could be specified upfront, as in that way only peaks within this range can be annotated.

4) Next to negative mode PGC data, also positive mode C18-orbitrap data with an 2-AB label was used to test the tool by the reviewer. Unfortunately, this did not result in any hits, while the manuscript indicated inclusion of this type of data in the tool training. It would be very valuable if other data types could also be analyzed with the current tool, but as this seems less functional at the moment, it should be accurately described in the manuscript for what data types the tool is working and for what data types not (yet). Also the title of the manuscript can be adjusted to be more specific, accordingly.

5) Minor issues encountered:

- a. For the plotting of the annotated spectra in step 4, the code had to be changed: plot_width/height into width/height, to prevent an error of not recognizing "plot_width" and "plot_height"
- b. The dimensions of the output on the webpage are not optimal for easy viewing, the alternative glycan structures appeared out of screen without a scrolling possibility.

6) Next to knowing what MS signals could be assigned to what glycan structures, other valuable information would be what MS/MS spectra were not assigned to a glycan and – more importantly – which of these do contain diagnostic glycan fragments? The latter would indicate unexpected glycoforms/monosaccharides/modifications. To not overlook these unexpected glycoforms it is essential that these are pointed out so they could be targeted for manual evaluation. It is unclear to what extent the tool provides this information.

7) The authors state "we set out to augment our pipeline to allow for, limited, zero-shot prediction outside our 3,508 defined glycans". It is unclear how automated and integrated this already is in the current tool.

8) The manuscript highlights the importance of active maintenance and updates for these type of tools and the fact that this is not done for other tools in the field. It is indeed a large frustration of researchers in the field that tools are often not completely functional or long-term supported. The long term value of the current tool will only be high when actively maintained and regularly updated with new data. How will the authors ensure this in a dynamic and demanding academic environment?

9) Especially for low-resolution iontrap data, m/z calibration can be particularly bad. Furthermore, PGC systems show often shifts in retention time. How does this affect the performance of the tool, is pre-calibration/RT alignment necessary? And would it improve annotations if accuracy could be identified upfront?

10) Currently, only options for trap data are provided in the user interface, what about data acquired on e.g. qTOF systems? Can these be processed, or will this be an option in the future?

11) It is already pointed out that "we do not currently support every type of glycan modification within

CandyCrunch and CandyCrumbs". Still, this would be highly valuable with newly emerging modifications and labels used in (functional) glycan analytics, as well as for the detection of uncommon monosaccharides not covered in the training data. How do the authors envision to enable these type of analysis in the future, possibly even without the training data available to incorporate these diverse modifications into the training models?

12) Finally, the visual appearance of especially Figure 1 should be improved: there is a large variation in font sizes used, panel C is hard to understand, panel D and F are very small. It is unclear what panel E is based on, can this be included in the figure? Panel H is very large and contains relatively little information (and what does the 9 mean?). Text in G is hardly readable (same for Fig 2A).

Author Rebuttal to Initial comments

Reviewer Comments: Reviewer #1 (Remarks to the Author):

There are serious concerns about this manuscript. It is well established that existing tandem MS methods do not suffice to assign complete glycan topologies. That glycan epitopes have diagnostic intensity ratios does not equate to the ability to assign the complete glycan structure. The mass spectral data do not contain sufficient information to define the glycan structure. It appears that the authors focus on differentiating glycan isomers using their algorithm. The challenge here is that different tandem MS modes produce different diagnostic ions. Considering data produced using a single tandem MS mode, there are considerable variations in dissociation conditions among data sets that make identification of consistent product ion ratios very difficult. In the protein folding field, AlphaFold succeeded because of the large set of high-quality protein three dimensional structures available to train the algorithm. By contrast, the glycomics field has yet to achieve consensus regarding how complete glycan structure assignments can be made from tandem mass spectra. The available data do not constitute high quality assignments necessary to train an algorithm. It is therefore not clear whether the CandyCrunch predictions are meaningful since they may reflect biases inherent in the raw data rather than structural determinants. To resolve these concerns, it is necessary for the authors to analyze new data on known glycans to illustrate the performance of the algorithm. These experiments must be performed blinded. The examples showing that complex N-glycome data can be annotated do not suffice since there is no ground truth. The authors must use ground truth samples to demonstrate the effectiveness of their algorithm.

We respectfully disagree with the assessment of the reviewer. It is not, and cannot be, proven that the task of assigning complete glycan topologies via tandem MS is impossible in principle. Especially given the fact that no substantive arguments or formal analyses are referenced, we do view this as the personal opinion of the reviewer (perhaps shared by a large fraction of the community, yet this still in no way constitutes a formal argument that can, or should, be used to discard advances). The fact that humans are unreliable in this effort does not mean at all that it is impossible in principle. Further, our training data to large extents include data for which the original annotators have used additional methods (e.g., exoglycosidase digestion, MS3, etc.) to refine annotations, which additionally strengthens the quality of our labels. As mentioned below, we have now also further improved the quality of our data by substantial amounts of manual data cleaning, improving model performance in the process.

The comparison to AlphaFold is telling indeed, as (i) the PDB is replete with factual errors (see, e.g., doi: 10.1111/febs.14320 or 10.1107/S2059798322011901) and (ii) AlphaFold does, in fact, reproduce many of the biases inherent in the raw data, such as not correctly modeling flexible regions etc. Its obvious success is only obvious in hindsight and was in all likelihood criticized beforehand as “not possible in principle” by many, as most breakthrough successes are. Yet it still is tremendously and undeniably useful, despite these shortcomings.

In fact, we would argue that, even in the proposed worst-case scenario (that we would “only” recapitulate the annotation capabilities and biases of human experts), we would still view our work as a major advance, as (i) by definition, this would constitute the current state-of-the-art, (ii) structural annotation is the current undisputed bottleneck in glycomics experiments, and (iii) we conclusively show that CandyCrunch can perform this feat multiple orders of magnitudes faster than humans.

The fact that different tandem MS modes produce different diagnostic ions is explicitly considered in this work, as CandyCrunch (unlike any other prediction software for this task) uses experimental set-

up data (such as ion mode and other characteristics) during prediction. We also train on the largest available dataset, which includes a substantial number of experimental set-ups. All this does not only mean that we explicitly account for differential fragmentation but also are more generalizable than any other approach to this problem. We further note that the coverage of experimental set-ups in our dataset, by definition, largely reflects their current usage and popularity within the scientific community. Further, a great advantage of deep learning models is that they can simply be re-trained once more/different data become available (such as ion mobility data instead of liquid chromatography), which would not be the case for bespoke, set-up specific methods.

We also reject the claim that we do not perform blinded experiments on ground truth data. Figure 1H, Supplementary Figure 6, Supplementary Figure 7, Supplementary Figure 9, Supplementary Figure 14, and Supplementary Table 4 are all reporting on various applications of CandyCrunch on (i) blinded data, (ii) with ground truth labels. All of these experiments conclusively demonstrate and reinforce the performance and utility of our presented model.

We further note that, even beyond our AI prediction model, our manuscript contains numerous independent advances (e.g., the largest curated set of glycomics MS2 data, a novel method to annotate MS2 fragments with CandyCrunch, data science approaches to obtain diagnostic ions, or the elucidation of fragmentation mechanisms for isomers via molecular dynamics simulations), none of which has been discussed or critiqued by the reviewer.

Reviewer #3 (Remarks to the Author):

Urban et al. present a deep learning model for the automated prediction of glycan structures from LC-MS/MS data. Such tools are seriously needed in the field, and are especially valuable when easy to use, providing accurate information about prediction certainty, and insight in the spectra annotation to allow expert evaluation. The tool presented here is highly promising in addressing these points. After reading the manuscript (clearly written, although the authors might have another look to simplify the text by using less complex wording) and trying the tool via the google drive-based interface, overall evaluation of it is positive, however the general feeling arises that the publication of the tool is slightly rushed and not all functionality is as good as envisioned/described in the text, also some unclarities and points for improvement remain:

1) The tool was evaluated by the reviewer on PGC-MS data (negative mode) of reduced N-, O- and GSL-glycans, as this was the main training data described in the manuscript. The tool was considered relatively easy to use, although multiple file name recognition errors were encountered which appeared to be resolved by refreshing the page multiple times. Also, while indicated that mzXML would be a suitable format, in our hands only mzML files could be run. While the tool was able to pick up large part of the glycans that were also manually annotated, some of them remained unidentified, especially observed for a couple of isomeric structures. On the other hand, some new, low abundant structures were found that were not picked up manually. Finally, some structures were reported that could not be found back in the data when evaluated manually. All in all this shows that the tool is valuable in the data analysis process and can be used as a first step in the data analysis workflow, however it remains necessary to manually/by another tool do a MS1 peak picking to see what potential glycan signals are not annotated (see comment 6) and to manually evaluate, especially the low scoring glycans.

While we have successfully tested our workflow also with mzXML files, we usually work with mzML files (which are the community-recommended format, in contrast to the obsolete mzXML format) and have tested these much more extensively. However, we have now engaged with testing more mzXML files and have improved our data loading. We note that the temporary file name recognition errors mentioned by the reviewer stem from the delay between uploading files on Google Drive and Google Colab recognizing said files. This is an unavoidable consequence of using Google Colab but we would like to note that our notebook on Google Colab is a complimentary addition to our work, while our main innovation is the method itself, which can also be used locally via our Python package. We are further enthusiastic to announce that we have substantially improved CandyCrunch as a model, achieving higher prediction performance, with an average structural accuracy of 90.3% on the validation dataset. This has been mostly achieved by more training data and cleaner training data, with great efforts of manual data cleaning. This new model and associated code improvements have been deployed in the version update 0.1.3 of our software package. We are convinced that these improvements make CandyCrunch more useful in practice, by delivering more reliable and complete predictions.

2) To enable this manual evaluation a couple of things are essential in the output of the tool, including the compositional annotation of the glycan often used in MS data, indicating the number of hexoses, hexnacs, fucoses etc in the glycan (eg. H4N5F1 or Hex4HexNAc5Fuc1), the theoretical mass of the annotated glycan and the ppm error between the data and the structure found. Importantly, it should be easy to look at the fragmentation spectrum used, including the annotated peaks for the glycan fragments (see comment 3).

We have added composition, theoretical mass, and ppm error to our output table to facilitate evaluation of predictions. We note however, that our nominal ppm errors might be higher than those achieved by commercial peak picking software, as we use an averaged spectrum for prediction (including isotope peaks, as they carry identical information in our binning scheme).

3) The current solution to evaluate the annotated fragmentation spectrum (CandyCrumbs) is not sufficient to allow an easy look at the data. Mainly because:

- a. Multiple peaks are annotated with the same fragments.
- b. You only see the fragments when you mouse-over a peak and often the content of the mouse-over is not readable because of some screen dimension issues.
- c. There is no indication of the ppm error between the theoretical fragment and the observed peak, making it very difficult to evaluate the trueness of an annotation. It would be good if the mass accuracy of the used method could be specified upfront, as in that way only peaks within this range can be annotated.

For point c, we have added the ppm error of theoretical fragment and observed peak. We also note that we already support this kind of filtering, as the *threshold* value can be altered to change the maximum permitted difference between theoretical and observed m/z . For point a, in some cases these multiple annotations can correspond to isotope peaks, which would indeed be the same glycan fragment. For point b, we would like to note that the notebook as a whole, but also the convenience of a fragment viewer, are not main parts of our manuscript and were hence not viewed as finished products but rather as platforms for initial users to experiment (given that users would need to upload their raw files to Google Drive, it was never intended as a mature solution; for that we advise users to run CandyCrunch locally).

4) Next to negative mode PGC data, also positive mode C18-orbitrap data with an 2-AB label was used to test the tool by the reviewer. Unfortunately, this did not result in any hits, while the manuscript indicated inclusion of this type of data in the tool training. It would be very valuable if other data types could also be analyzed with the current tool, but as this seems less functional at the moment, it should be accurately described in the manuscript for what data types the tool is working and for what data types not (yet). Also the title of the manuscript can be adjusted to be more specific, accordingly.

We thank the reviewer for spotting this. CandyCrunch as a model can certainly use and predict 2-AB labelled glycans. However, in our post-prediction workflow used in the notebook, when we check whether predictions have the correct mass, we did not consider tags except for reduction. Thus, all predictions had the “wrong” mass and were discarded. We have now remedied this by allowing for an optional *mass_tag* argument in our prediction function, supporting all tags for which we have training data and also allowing custom masses beyond that.

5) Minor issues encountered: a. For the plotting of the annotated spectra in step 4, the code had to be changed: *plot_width/height* into *width/height*, to prevent an error of not recognizing “*plot_width*” and “*plot_height*” b. The dimensions of the output on the webpage are not optimal for easy viewing, the alternative glycan structures appeared out of screen without a scrolling possibility.

This error has been caused by an update of the Bokeh package on Google Colab after our initial submission. We fixed this error at the end of August and have now also exactly specified the version of Bokeh within the notebook, to prevent such a change in the future.

6) Next to knowing what MS signals could be assigned to what glycan structures, other valuable information would be what MS/MS spectra were not assigned to a glycan and – more importantly – which of these do contain diagnostic glycan fragments? The latter would indicate unexpected glycoforms/monosaccharides/modifications. To not overlook these unexpected glycoforms it is essential that these are pointed out so they could be targeted for manual evaluation. It is unclear to what extent the tool provides this information.

This exact functionality is already possible when setting *get_missing* to True in our prediction function. The only limitations here are that spectra need to still pass our filters (e.g., diagnostic glycan fragments, minimum number of peaks, etc.) in order to be retained. We have made this clearer in our revised manuscript.

7) The authors state “we set out to augment our pipeline to allow for, limited, zero-shot prediction outside our 3,508 defined glycans”. It is unclear how automated and integrated this already is in the current tool.

Our Python package CandyCrunch is fully able to use the mentioned zero-shot workflows. However, for our public-facing Google Colab notebook we have chosen to make these workflows opt-in, because they tend to be more error-prone / require more expert review to properly evaluate. We view these procedures as advanced use cases of CandyCrunch. We now make this distinction clear in the revised manuscript. Parenthetically, we note here that our new glycan number (3,391) stems from our data cleaning efforts, that have for instance reduced unnecessary label uncertainty (such as converting Fuc(?1-?) into Fuc(a1-?), reducing the number of unique glycans).

8) The manuscript highlights the importance of active maintenance and updates for these type of tools and the fact that this is not done for other tools in the field. It is indeed a large frustration of researchers in the field that tools are often not completely functional or long-term supported. The

long term value of the current tool will only be high when actively maintained and regularly updated with new data. How will the authors ensure this in a dynamic and demanding academic environment?

While no firm guarantee can be given for this point, we have a proven track record of maintaining software (our glycowork package has been continuously updated and maintained since June 2021). Further, since we use CandyCrunch in our own research (and our core facility also has started to use this software), we have a natural interest to maintain and further develop it. We have in fact just received funding from the IngaBritt och Arne Lundbergs Forskningsstiftelse to further develop CandyCrunch, assuring long-term maintenance for the foreseeable future. We also note that we have updated the CandyCrunch package several times since the initial launch and are already gathering GitHub commits for a version update 0.2.0. Finally, by making all the code, models, and data open-source and publicly accessible, we substantially increase the chances of maintenance, especially if the community values this resource.

9) Especially for low-resolution iontrap data, m/z calibration can be particularly bad. Furthermore, PGC systems show often shifts in retention time. How does this affect the performance of the tool, is pre-calibration/RT alignment necessary? And would it improve annotations if accuracy could be identified upfront?

We have performed additional experiments to indicate that higher m/z resolution does not necessarily translate to higher prediction performance (new Supplementary Table 3). We hypothesize that is because of a greater separation of potentially confusable peaks than the typical m/z error (i.e., there are usually no other realistically possible peaks within X ppm around the "true" peak and thus their misannotation is not a widespread problem; see the new Supplementary Fig. 1). We also note that our used m/z remainder approach overrides binning resolution and allows us to recover exact masses as long as only one fragment is contained in a bin (which Supplementary Fig. 1 supports). Regarding retention time, we internally normalize retention times to relative times, as described in our manuscript, which partly remedies shifts in absolute retention time. However, we cannot exclude that RT alignment would further improve model performance.

10) Currently, only options for trap data are provided in the user interface, what about data acquired on e.g. qTOF systems? Can these be processed, or will this be an option in the future?

Data acquired on qTOF systems can be processed and simply requires users to select "other" on the trap selection. For improved clarity, we have now displayed all available options in the dropdown menu.

11) It is already pointed out that "we do not currently support every type of glycan modification within CandyCrunch and CandyCrumbs". Still, this would be highly valuable with newly emerging modifications and labels used in (functional) glycan analytics, as well as for the detection of uncommon monosaccharides not covered in the training data. How do the authors envision to enable these type of analysis in the future, possibly even without the training data available to incorporate these diverse modifications into the training models?

Several answers to this comment are possible, since this is a multifaceted problem, and we have added this consideration into our revised manuscript. We believe that at least a sizable fraction of all possible modifications and labels can be included into our workflow and describe several scenarios in which this holds true:

- Should new modifications/labels emerge and corresponding data are deposited, we (and any other researcher with modest computational capacities) can re-train CandyCrunch, facilitating

the analysis of this modification/label in other datasets. We have now further added a training script for CandyCrunch to our GitHub repository, maximizing reproducibility.

- For labels at strictly one position (such as the reducing end), a substantial number of fragments will still present without the label and can thus be used by CandyCrunch for prediction. Due to the sparsity of binned spectra, CandyCrunch relies much more on the presence of peaks, as a source of information, rather than their absence. While we still anticipate a performance decrease in these cases, we believe this approach should be feasible in principle.
- A wholly different approach would be to conceptualize glycans as chemicals and train models on the atom-level. This would in principle allow for an extension to chemically modified glycans, if general principles are learned by the model. However, we view this as an extensive endeavor and outside the scope of this work. It provides, however, a clear path for future research to improve and generalize models.
- Lastly, we note that CandyCrunch can be easily extended to new modifications/labels, even in the absence of training data, as long as the masses of the fragments can be calculated precisely (i.e., as long as that information is available).

12) Finally, the visual appearance of especially Figure 1 should be improved: there is a large variation in font sizes used, panel C is hard to understand, panel D and F are very small. It is unclear what panel E is based on, can this be included in the figure? Panel H is very large and contains relatively little information (and what does the 9 mean?). Text in G is hardly readable (same for Fig 2A).

We have improved the visual appearance of this figure. We also note that panel E is based on evaluation of the trained CandyCrunch model on our independent validation set (for instance available on Zenodo), after splitting the dataset into train and validation set. We describe this in the figure legend and now also refer to the Zenodo dataset. The "9" in panel H refers to the glycans not identified by either method.

Decision Letter, first revision:

Dear Daniel,

Thank you for your letter detailing how you would respond to the reviewer concerns regarding your Article, "Predicting glycan structure from tandem mass spectrometry via deep learning". We have gone over your revision plan and invite you to revise your manuscript as you have outlined. We agree with all the points that you mentioned are beyond the scope of current paper, and are happy to see that you plan to provide command line functionality for your software.

[Redacted]

We hope to receive your revised paper within 8 weeks. If you cannot send it within this time, please let us know. In this event, we will still be happy to reconsider your paper at a later date so long as nothing similar has been accepted for publication at Nature Methods or published elsewhere.

OPEN SCIENCE REQUIREMENTS

REPORTING SUMMARY AND EDITORIAL POLICY CHECKLISTS

Reporting summary: <https://www.nature.com/documents/nr-reporting-summary.zip>
Editorial policy checklist: <https://www.nature.com/documents/nr-editorial-policy-checklist.zip>

IMAGE INTEGRITY

When submitting the revised version of your manuscript, please pay close attention to our Digital Image Integrity Guidelines and to the following points below:

DATA AVAILABILITY

CODE AVAILABILITY

Please include a "Code Availability" subsection in the Online Methods which details how your custom

code is made available. Only in rare cases (where code is not central to the main conclusions of the paper) is the statement "available upon request" allowed (and reasons should be specified).

For more information on our code sharing policy and requirements, please see:
<https://www.nature.com/nature-research/editorial-policies/reporting-standards#availability-of-computer-code>

MATERIALS AVAILABILITY

SUPPLEMENTARY PROTOCOL

To help facilitate reproducibility and uptake of your method, we ask you to prepare a step-by-step Supplementary Protocol for the method described in this paper. We encourage authors to share their step-by-step experimental protocols on a protocol sharing platform of their choice and report the protocol DOI in the reference list. Nature Portfolio's Protocol Exchange is a free-to-use and open resource for protocols; protocols deposited in Protocol Exchange are citable and can be linked from the published article. More details can found at www.nature.com/protocolexchange/about.

ORCID

Nature Methods is committed to improving transparency in authorship. As part of our efforts in this direction, we are now requesting that all authors identified as 'corresponding author' on published papers create and link their Open Researcher and Contributor Identifier (ORCID) with their account on the Manuscript Tracking System (MTS), prior to acceptance. This applies to primary research papers only. ORCID helps the scientific community achieve unambiguous attribution of all scholarly contributions. You can create and link your ORCID from the home page of the MTS by clicking on 'Modify my Springer Nature account'. For more information please visit please visit www.springernature.com/orcid.

Sincerely,
Arunima

Arunima Singh, Ph.D.
Senior Editor
Nature Methods

Reviewers' Comments:

Reviewer #1:

Remarks to the Author:

At its core, CandyCrunch is still a spectral matching method. To evaluate the correctness of the predicted structure, CandyCrunch compares the experimentally observed fragmentation pattern with that predicted by AI from many tandem mass spectra of the given structure in the training dataset. Effectively, it utilizes a "consensus" spectrum rather than individual reference spectrum, leading to a (presumably) improved accuracy of top-ranked predictions (around 90%). The experimental parameters (ionization mode, LC types, glycan modifications, MS instrument types, glycan classes, etc.) are embedded in the CandyCrunch model architecture, allowing it to (somewhat) overcome significant variations in experimental conditions among data sets. The (normalized) retention time information and precursor m/z were also utilized by CandyCrunch to aid prediction. CandyCrunch further utilizes domain knowledge and biosynthetic knowledge to filter and refine predictions. Although it is conceptually infeasible to predict structures absent from the training set, the authors demonstrated limited subset zero-shot predictions utilizing the biosynthetic network. As the authors pointed out in their rebuttal letter, the manuscript also contains several independent advances. For example, the curated set of glycomics MS2 data would be a valuable resource; CandyCrunch is a useful tool for spectral annotation; and MD simulations that elucidates fragmentation mechanisms. Although these are valuable, they, on their own, do not warrant publication on Nature Methods. The critique below will focus on the core advance of this study, namely the development of an AI prediction model, but would also comment on the values and flaws of other aspects of the manuscript when applicable.

1. A true zero-shot prediction should be made with synthetic glycan structures that are not present in the training dataset. However, this is not likely to succeed with the present approach that relies on the identification of existing structures in the sample set and biosynthetic network, especially for: (a) synthetic glycan structures that do not conform to known biosynthetic knowledge; and/or (b) identification of individual glycan structure that may or may not be biologically relevant, from an isolated tandem MS spectrum. An alternative approach to demonstrate the limited feasibility of zero-shot prediction is to purposely remove all tandem mass spectra of selected structures from the training set, that is, perform the training/test split at the structure level rather than the sample level, and see if CandyCrunch can recover these structures using the present approach. This would constitute a better testing strategy on the performance of zero-shot prediction, as the structures removed are validated structures rather than tenuous assignment (that are likely made on low-quality spectra or low-abundance species). The authors should be able to perform this evaluation with the available data, and if successful, could significantly strengthen the manuscript.
2. CandyCrunch utilizes domain and biosynthetic knowledge, as well as retention time information to (presumably) improve the prediction accuracy, which underlined the limitations of utilizing only

fragmentation pattern for structure prediction. This is not surprising because: (1) biosynthetic knowledge would reduce the number of possible structures for a given mass by many orders of magnitude; and (2) many isomers have similar fragmentation patterns, especially when acquired by CID in ion trap instruments. The authors should provide a quantitative evaluation of how the application of each filter/refining process improves the prediction accuracy (e.g. prediction accuracy with only the MS₂ spectra; with the addition of retention time; with the addition of the biosynthetic knowledge; etc.)

3. According to the authors, "even erroneous predictions are structurally close to the correct solution". However, "closeness" is often insufficient to answer many biological questions. In fact, even a slight change in one structural variable can have considerable impact on the biological function of a glycan. A better indication of the evaluation accuracy would be the confidence score. It is advisable for the authors to develop a confidence score to indicate the likelihood of correct assignment. Once this is established, the authors can present a statistical distribution of the confidence score for correct assignments, and for incorrect assignments. This is important to inform the analyst how much they should trust the result.

4. The authors stated CandyCrunch is suitable for analysis of glycans with common modifications, such as permethylation. It would be useful for the authors to provide a statistical evaluation of the prediction accuracy utilizing different experimental approaches (beyond the statement that negative-mode CID of reduced glycans tends to produce the best results).

5. How would CandyCrunch perform on tandem mass spectra of mixture? Co-elution is common for isomeric glycans. Would it be able to identify the presence of multiple structures in a single chimeric spectrum?

6. Retention time can vary significantly from one data set to another, especially for PGC-LC analysis. The authors perform retention time normalization by dividing the absolute retention time by the respective maximal retention time. I do not think this is the correct way to normalize retention time. The retention time should be normalized using the glucose unit (dextran ladder) or at the worst-case scenario, against known structures present in the sample.

7. The authors also performed grouping of averaged spectra in chunks of 0.5-minute retention time window, but many isomers would co-elute in that time frame.

8. Biosynthetic network analysis would only be feasible if the entire glycome is analyzed. What would be the strategy for analysis of a single isolated structure?

9. The authors stated that they also checked missed Neu5Gc-substituted Neu5Ac-glycans and vice versa with a mass shift of 16 Da and corresponding diagnostic ions. Please comment on if this check extends beyond single substitution.

10. The authors noted that "analyzing the data at higher resolution does not give rise to higher accuracy". The simulation of higher resolution data was performed by increasing the number of bins (Supporting Table 3), leading to an increased "effective resolution per bin". They further stated that the fact that they "used m/z remainder approach overrides binning resolution and allows us to recover exact masses as long as only one fragment is contained in a bin (which Supplementary Fig. 1 supports)". This is inaccurate. Without a sufficiently high resolution, it would be impossible to differentiate isobaric fragments (e.g. those differ in composition of CH₄ vs. O): no binning can improve the resolution of the original data (at best, it can preserve the resolution), and no m/z remainder approach can improve the mass accuracy of the original data. The true test should be done by utilizing the subset of real high-resolution data obtained on TOF or Orbitrap instruments, and compare the performance. Additionally, Supplementary Figure 1 is suspect, as it only considers glycosidic bond cleavages. Cross-ring cleavages and neutral losses are common in glycan tandem mass spectra. Figure S1 does not reflect the true extent of potential occurrence of multiple fragments in a single bin.

11. The authors noted that “a model only trained on O-glycans performed worse for predicting O-glycans (Supplementary Table 4; topology: 84% accuracy, structure: 79% accuracy) than the model trained on all classes”, and hypothesized that “this was due to the structure-based loss function we used for training, as well as shared information between spectra of different classes, stemming from shared glycan motifs across classes”. This is an interesting point. How is the motif information incorporated into the training algorithm? Would CandyCrunch be able to predict structures that are built from existing motif but not in the training set? Again, this could be tested by removing certain structures entirely from the training set.

12. CandyCrunch is a useful tool, but the way it prioritizes assignments when multiple assignments are possible is suspect (Figure S10). There are often cases where a neutral loss or double cleavages are more abundant than a single-cleavage fragment. Is it possible to also learn the assignment from the annotated training set?

Overall, I think the manuscript is worth consideration for publication on Nature Methods, but there are serious flaws that need to be addressed and it needs further and more robust evaluation.

Reviewer #3:

Remarks to the Author:

The rebuttal addresses all my comments well and the authors updated and improved the manuscript. The described tool has a lot of options to tune it for a specific experiment and train it with new data, and it will likely be very valuable in LC-MS based glycomics experiments in the future.

Although the authors provided an easy accessible Google Drive interface to make the tool accessible for non-python-specialists, this is not an ideal interface to bring the tool to its full potential (as indicated by the authors themselves as well). I understand that this is not the main purpose of the manuscript, but I would like to encourage the developers to equip the Python tool with a user friendly interface to make the tool accessible for all glyco LC-MS specialist, independent of their programming skills. As a start, a clearer “read me” can be provided on how to get started with the tool using the local installation. I’m sure there will be enough “dummy testers” in the field happy to help validating this.

Reviewer #4:

Remarks to the Author:

This manuscript presents a highly novel tool for glycomics data annotation which is made available via an ad hoc Python notebook. The authors have addressed both the novelties of the software as well as the drawbacks or points for users to be aware of when using the software. In particular, it is very important to make potential users aware of what NOT to expect.

The data and methodology is rigorously explained, and the manuscript is written clearly overall.

Suggestions for improvement:

1. A table or at least a description of the various options that are available could be summarized. While the author responded to one of the other reviewer's comments and added a statement that the given option (zero-shot) is available, amongst the comments to the reviewers there seemed to be many such options that are available, but the user would not be aware of them. So it would help the manuscript greatly to indicate that all of the various options that often need to be considered when annotation glycomics spectra are addressed, and the list would indicate how to set those options.

2. While users could use the outputted annotations as figures to include in their own publications, how do the authors consider feeding back the annotations to databases such as in the GlySpace Alliance? In particular, are GlyTouCan IDs assignable to the annotations?

Reviewer #5:

Remarks to the Author:

The manuscript by Urban et al. describes a method for identification of glycan structures from glycomics mass spectrometry data using deep learning. The method clearly provides a dramatic improvement in capability vs existing software methods, both in performance and generalizability to a variety of experimental and instrumental setups. This is both a very challenging and very necessary endeavor, making the manuscript likely of great interest to a broad glycoscience community. The manuscript is well written and, particularly following the revisions from earlier reviews, clearly describes the method, results, and current limitations. I have some specific comments on the method and results below, but most are very minor issues.

The only major issue I see is the requirement that the software be accessed programmatically in Python, with no provision of any user interface or command line accessibility beyond the CoLab notebook demo. While this is not an issue for experienced programmers, to truly “democratize structural glycomics” beyond a few specialists, this represents a major roadblock for glycobiochemists and other glycoscientists who may not have the programming skills (or time) to write their own code to use the method. The authors refer to the CoLab demo as not a main part of the manuscript in responding to the initial review comments about its limited functionality, but these comments are a clear indication of the need and desire for a functional interface to fully realize the potential of the method. This dovetails with concerns about the method receiving long term support and engagement – if the barrier to use is too high and few people adopt the method, there is less feedback and external motivation to keep maintaining and improving it. I would urge the authors to provide at least a minimal command line execution option (e.g., pass in a config file that has the parameters and data paths) to run the analysis and generate at least a basic output table or report. Further development (e.g., of a graphical interface) could come later or as warranted from community feedback, but I think there needs to be some provision to run the software without having to write Python code for it to be considered a community resource as opposed to a specialist tool for bioinformaticians. That said, the CoLab demo and source code clearly show a high level of programming expertise, so I have confidence in the authors’ capability to do this.

Minor Comments:

- “Applied to fully unseen datasets, CandyCrunch routinely achieved high performance (Supplementary Table 4; topology: 92% accuracy, structure: 84%)”. These numbers appear to be Top5 structure accuracies from Supp. Table 4, whereas the previous data in Figure 1 was presented as Top1 accuracy. This needs to be noted, and ideally, the range of Top1 accuracies should be provided as well or instead.
- With regard to Supp. Table 3, the authors state that higher m/z resolution appeared not to necessarily translate to higher prediction performance. The resolutions tested (0.7 to 1.4 Da per bin) are 2 orders of magnitude lower than high resolution mass spectrometry data, so it would be worth

noting the difference between “higher” resolution than the model default (i.e., 2x higher) and “high” resolution in a mass spectrometry context (i.e., 100x higher).

Author Rebuttal, first revision:

We thank all reviewers for their insightful comments and suggestions for improvement. We have fully addressed these comments in our substantially revised manuscript by engaging in extensive new analyses, leading to new supplemental figures and tables, as well as numerous text modifications and additions, and improvements to our Python package and beyond. In summary, we have (i) demonstrated zero-shot prediction capabilities by purposefully removing structures from the training data (**new Fig. S8**), (ii) added the option of using CandyCrunch via a command line interface (**new CandyCrunch v0.3**), (iii) performed ablation experiments to illustrate how much of our performance comes from different sources of information (MS2 spectrum, retention time, etc.) (**new Table S3**), (iv) demonstrated that correctly predicted structures, on average, exhibit higher confidence scores (**new Table S7**), (v) added subgroup analyses describing the performance of CandyCrunch on various glycan modifications (e.g., permethylation) (**new Table S5**) as well as high-resolution data (e.g., orbitrap) (**new Table S6**), (vi) added a multi-sample workflow that aligns retention times across samples from the same experiment and accommodates shifts in retention time (**new Fig. S9**), (vii) provided more options to customize retention time precision when using CandyCrunch (e.g., minimum/maximum cut-offs, time window resolution) (**new CandyCrunch v0.3** and **new Table S12**), (viii) extended the documentation of CandyCrunch, with particular emphasis on available options and how to fill them, and a guide on how to work with the software when installed locally (**new Fig. S2, new Table S12, new CandyCrunch v0.3**), (ix) expanded our strategy of searching for more extensively Neu5Gc-/Neu5Ac-substituted glycans, as well as GlcNAc-/GlcNAc6S-substituted O-glycans, to increase zero-shot capabilities (**new CandyCrunch v0.3**), (x) documented the inherent cross-training with our approach by showing that glycans with shared motifs need fewer training spectra to be correctly captured by the model (**new Fig. S6**), (xi) added GlyTouCan IDs as a new column in our output table, whenever available (**new CandyCrunch v0.3**), and (xii) demonstrated that our approach is robust to retention time overlaps (**new Fig. S3**). Changes in the manuscript and point-by-point responses here are colored in blue. We believe that these changes have substantially improved our manuscript, contextualized our findings, and will allow readers to better evaluate our analyses and findings.

Reviewer #1:

Remarks to the Author:

At its core, CandyCrunch is still a spectral matching method. To evaluate the correctness of the

predicted structure, CandyCrunch compares the experimentally observed fragmentation pattern with that predicted by AI from many tandem mass spectra of the given structure in the training dataset. Effectively, it utilizes a “consensus” spectrum rather than individual reference spectrum, leading to a (presumably) improved accuracy of top-ranked predictions (around 90%). The experimental parameters (ionization mode, LC types, glycan modifications, MS instrument types, glycan classes, etc.) are embedded in the CandyCrunch model architecture, allowing it to (somewhat) overcome significant variations in experimental conditions among data sets. The (normalized) retention time information and precursor m/z were also utilized by CandyCrunch to aid prediction. CandyCrunch further utilizes domain knowledge and biosynthetic knowledge to filter and refine predictions. Although it is conceptually infeasible to predict structures absent from the training set, the authors demonstrated limited subset zero-shot predictions utilizing the biosynthetic network. As the authors pointed out in their rebuttal letter, the manuscript also contains several independent advances. For example, the curated set of glycomics MS2 data would be a valuable resource; CandyCrunch is a useful tool for spectral annotation; and MD simulations that elucidates fragmentation mechanisms. Although these are valuable, they, on their own, do not warrant publication on Nature Methods. The critique below will focus on the core advance of this study, namely the development of an AI prediction model, but would also comment on the values and flaws of other aspects of the manuscript when applicable.

We thank the reviewer for engaging with our work and providing us with the opportunity to improve it. Below, we have addressed each of the individual points and describe how this has improved our manuscript.

1. A true zero-shot prediction should be made with synthetic glycan structures that are not present in the training dataset. However, this is not likely to succeed with the present approach that relies on the identification of existing structures in the sample set and biosynthetic network, especially for: (a) synthetic glycan structures that do not conform to known biosynthetic knowledge; and/or (b) identification of individual glycan structure that may or may not be biologically relevant, from an isolated tandem MS spectrum. An alternative approach to demonstrate the limited feasibility of zero-shot prediction is to purposely remove all tandem mass spectra of selected structures from the training set, that is, perform the training/test split at the structure level rather than the sample level, and see if CandyCrunch can recover these structures using the present approach. This would constitute a better testing strategy on the performance of zero-shot prediction, as the structures removed are validated structures rather than tenuous assignment (that are likely made on low-quality spectra or low-abundance species). The authors should be able to perform this evaluation with the available data, and if successful, could significantly strengthen the manuscript.

We first would like to respectfully challenge this with our view that zero-shot prediction is a niche case in the current practice of glycomics, especially given our extensive prediction repertoire of ~3,400 unique structures. The most common use case, especially in the realm of high-throughput studies, where CandyCrunch would reap the greatest benefits, are samples which

do not exhibit structures that would lie outside our ~3,400 structures (e.g., serum glycans etc.). Very few people work on the characterization of highly novel structures found in exotic species and this cannot be viewed as a common use case under any definition. Further, we would make the conjecture that few, if any, humans would have experience in annotating ~3,400 structures, such that many zero-shot annotations for any given human would not be thus for our model.

However, we acknowledge the point that we should objectively demonstrate the capability of our additional workflows to engage in proof-of-concept zero-shot predictions, to substantiate our claim. For this purpose, we have added new analyses, in which – as suggested by the reviewer – we have removed all spectra of individual structures from the training set and measured whether these structures are captured via zero-shot predictions. This indeed worked robustly for the structures we analyzed, both for biosynthetic modeling as well as database inference approaches, the two main zero-shot approaches we employ in this work. The results from this analysis can be found in the **new Supplementary Fig. 8**.

2. CandyCrunch utilizes domain and biosynthetic knowledge, as well as retention time information to (presumably) improve the prediction accuracy, which underlined the limitations of utilizing only fragmentation pattern for structure prediction. This is not surprising because: (1) biosynthetic knowledge would reduce the number of possible structures for a given mass by many orders of magnitude; and (2) many isomers have similar fragmentation patterns, especially when acquired by CID in ion trap instruments. The authors should provide a quantitative evaluation of how the application of each filter/refining process improves the prediction accuracy (e.g. prediction accuracy with only the MS2 spectra; with the addition of retention time; with the addition of the biosynthetic knowledge; etc.)

This is a very important observation and we agree with the reviewer. We have now added the **new Supplementary Table 3** to show the results of the suggested ablation experiment on model performance. While we do see decreases in performance when removing certain modes of information, we note that (i) binned intensities alone are responsible for a large fraction of the performance and (ii) there is redundancy between information channels (e.g., one could imagine that, for an AI model, the information contained in precursor mass could be mostly reconstructable from MS2 information). Since neither computational resources, inference time, nor the acquisition of the required information are current bottlenecks in any sense, we opt to retain everything that gives us a boost in performance, no matter how minuscule. Importantly, the model using all types of provided information yields the highest prediction performance.

3. According to the authors, “even erroneous predictions are structurally close to the correct solution”. However, “closeness” is often insufficient to answer many biological questions. In fact, even a slight change in one structural variable can have considerable impact on the biological function of a glycan. A better indication of the evaluation accuracy would be the confidence score. It is advisable for the authors to develop a confidence score to indicate the likelihood of

correct assignment. Once this is established, the authors can present a statistical distribution of the confidence score for correct assignments, and for incorrect assignments. This is important to inform the analyst how much they should trust the result.

We agree that precision is important when evaluating biological functions. We still maintain, however, that it constitutes an unequivocal improvement to have errors, if they do occur, be closer to the truth. This is not something to be taken for granted, especially in machine learning, in which errors often strongly deviate from human intuition, and is one of the many aspects that distinguishes our work from those of others. Further, the preponderance of structural ambiguities (e.g., “?” or “HexNAc”) present in nearly all published glycomics datasets showcases that biological insights can be gleaned even in the absence of structural perfection.

We also note that the expectation of a reliable confidence score outstrips the current expectations the field has for human analysts, for whom no measures of confidence need to be specified. Further, CandyCrunch already produces a confidence score—the predicted probability of the final structure—and this is present in the user output and used within the workflow, for instance to filter out predictions with too low confidence values. What we cannot provide—and what would be an unrealistic expectation, also with respect to other machine learning applications—is a single, universal cut-off value, above which a user can trust the prediction. The main reason for this lies in combinatorics and probability density functions: larger glycans, even if correctly predicted, will always have smaller probability scores, on average. That is because the probability density is stretched across all possible isomers, dampening the apex of this function (which should correspond to the correct prediction).

What we can do, however, to provide some guidance, is to compare the confidence scores of the same structure for correct and incorrect top1 predictions. While this will not result in a universal cut-off, it at least shows that, on average and controlling for the combinatorial potential of a specific glycan composition, correct predictions showcase statistically significant higher predicted probabilities than incorrect predictions. In fact, ~94% of structures exhibit a higher confidence score if they are indeed the correct prediction (~78% even reaching statistical significance after stringent multiple testing correction). This is, in part, ensured by the prediction confidence calibration that we perform via Platt scaling. We demonstrate this in the **new Supplementary Table 7**.

4. The authors stated CandyCrunch is suitable for analysis of glycans with common modifications, such as permethylation. It would be useful for the authors to provide a statistical evaluation of the prediction accuracy utilizing different experimental approaches (beyond the statement that negative-mode CID of reduced glycans tends to produce the best results).

We agree with the reviewer and have now added a new supplemental table detailing model performance for modifications for which we have sufficient amounts of data (**new Supplementary Table 5**). We note that, as for any machine learning effort, our model will

always perform best, on average, with the settings that contributed the most data. We added appropriate cautionary notes to that effect into the revised manuscript.

5. How would CandyCrunch perform on tandem mass spectra of mixture? Co-elution is common for isomeric glycans. Would it be able to identify the presence of multiple structures in a single chimeric spectrum?

In principle, this can be detected by CandyCrunch, though we caution that this would also be substantially harder for human analysts, compared to non-chimeric spectra. Since the model ranks structures by probability, the situation can arise that multiple structures all have reasonably high predicted probabilities. We also always output the top ranked structures in our standard prediction output, so this information is inherently available to users. We note though, that in the column denoting the “final” prediction we, by default, always have a single structure. In the mentioned case, this would then result in the more dominant structure being chosen for this column (we assume here that an exact 50/50 mixture is a rare exception). As further discussed below, as long as there is no perfect overlap in co-elution (i.e., the peaks do not exactly and completely overlap), we still routinely detect both isomers in the output table, with the minor caveat that we might have some inaccuracies in the relative quantification of the respective isomers, due to overlap. We raise these important points in the revised manuscript.

We also tested this empirically by searching for isomers eluting with a low mean retention time difference. We first would like to note that we had major difficulties of finding cases like this in our dataset of ~500,000 MS2 spectra, either indicating that this does not seem prevalent or that human experts do not commonly annotate co-eluting structures as well. One of the few examples we identified that fulfilled at least some criteria (while not a strict isomer, the m/z difference was below 0.5 and thus below our default cut-off of 0.5, to establish precursor ion identity, making the structures isobars for our model) can be found in the **new Supplementary Fig. 3**. Herein, we show that, despite overlaps in retention time, CandyCrunch is able to confidently assign both structures, including in the overlapping range.

6. Retention time can vary significantly from one data set to another, especially for PGC-LC analysis. The authors perform retention time normalization by dividing the absolute retention time by the respective maximal retention time. I do not think this is the correct way to normalize retention time. The retention time should be normalized using the glucose unit (dextran ladder) or at the worst-case scenario, against known structures present in the sample.

We concur that retention time variation is an issue and concede that our approach to normalize retention time is inferior to absolute normalization methods, such as via glucose units. However, this approach is currently infeasible because we simply do not have this information for our training data. It would also not work to only use glucose unit-normalized retention times at the time of prediction, if the model has not been trained on similar data. Similarly, normalization

against known structures would lead to varying retention time associations across samples, contingent on whether a certain structure is present or not. We also would like to note that, while such approaches might further improve model performance, our currently reported model performance, achieved with our current retention time approach, is substantially above alternative methods.

However, since we acknowledge the importance of retention time variation, we have developed a new workflow for using CandyCrunch with a batch of samples. In this workflow, next to our already described advances, we align retention times across samples, if possible and suitable, to build a prediction library and ensure that shifts in retention time between samples are accommodated. In the **new Supplementary Fig. 9**, we demonstrate that this is a useful addition to further improve and harmonize predictions. Of course, we do acknowledge that this does not entirely cover the concern of the reviewer, as it would, for now, only apply to multiple samples of the same experiment, yet we still view it as a meaningful advance toward this purpose and are confident that further developments will make this even more potent.

This analysis, or rather the evaluation of it, also highlighted the occasional presence of single outlier spectra (e.g., m/z 384 close to 40 minutes in a 45-minute LC run, yet still containing glycan fragments such as m/z 204) that were discarded by the original analyst and thus present a large retention time difference in our evaluation. In case users wish to avoid the occurrence of these spectra in the output, we have used this opportunity to showcase three solutions: (i) the abovementioned multi-file workflow, which remedies these outliers, (ii) setting the new “rt_max” keyword argument to avoid investigating the end of the LC run, or (iii) using “experimental = True”, as we have now added a new preliminary step of removing retention time outliers to this part of the workflow (**new v0.3** of CandyCrunch). We expect that the last option, after more careful and comprehensive testing, will eventually make its way into being always applied within *wrap_inference*.

7. The authors also performed grouping of averaged spectra in chunks of 0.5-minute retention time window, but many isomers would co-elute in that time frame.

We acknowledge the concern of the reviewer and have now added the option in the CandyCrunch software for users to more precisely specify the treatment of retention time (**new v0.3** of CandyCrunch). This includes the settings of minimum and maximum cut-offs, avoiding potential false-positives of predicting glycans at very high retention times, as well as the mentioned value for retention time windows.

We still would like to clarify that this is only a concrete problem for CandyCrunch if isomers co-elute in a perfect manner. Partial co-elution is not an inherent problem. As long as there is a 0.5 minute window that predominantly contains one isomer (even if the spectra would be technically chimeric and not from the central part of the peak), predictions would reflect both isomers, which would survive our deduplication efforts, and be contained within our annotation output table. As

mentioned above, at worst this will moderately affect the estimated abundance of each isomer. We here would again like to point to our **new Supplementary Fig. 3**, to substantiate these arguments.

The empirical support of our dataset of 500,000 MS2 spectra, indicating a paucity of isomers with mean retention time differences below 0.5 minutes, supports our view that co-elution is predominantly partial and thus not a fundamental problem for our approach.

8. Biosynthetic network analysis would only be feasible if the entire glycome is analyzed. What would be the strategy for analysis of a single isolated structure?

It is correct that biosynthetic network analysis would be inapplicable to single isolated structures. In that case, this additional analysis would just not contribute anything to the annotation, though the rest of our workflow would still normally apply. We have added this consideration to the revised manuscript. We note that biosynthetic network analysis in its current form is a supplemental analysis and also optional within the workflow.

9. The authors stated that they also checked missed Neu5Gc-substituted Neu5Ac-glycans and vice versa with a mass shift of 16 Da and corresponding diagnostic ions. Please comment on if this check extends beyond single substitution.

Previously, this substitution related to replacing a single Neu5Ac with Neu5Gc (or vice versa). In the revised manuscript and CandyCrunch package (**new v0.3** onward), we now replace up to all of Neu5Ac with Neu5Gc (or vice versa; i.e., all possible options), to have a broader coverage. This was the only check of that sort at the time. One could envision to extend this to other similar scenarios (such as checking for otherwise identical O-glycans with/without a GlcNAc6S modification) and we have now made the first step toward this by adding a step to check/impute for sequences with GlcNAc/GlcNAc6S connected to the reducing end GalNAc (**new v0.3** of CandyCrunch). We will explore this approach further in the future and thank the reviewer for encouraging this improvement.

10. The authors noted that “analyzing the data at higher resolution does not give rise to higher accuracy”. The simulation of higher resolution data was performed by increasing the number of bins (Supporting Table 3), leading to an increased “effective resolution per bin”. They further stated that the fact that they “used m/z remainder approach overrides binning resolution and allows us to recover exact masses as long as only one fragment is contained in a bin (which Supplementary Fig. 1 supports)”. This is inaccurate. Without a sufficiently high resolution, it would be impossible to differentiate isobaric fragments (e.g. those differ in composition of CH4 vs. O): no binning can improve the resolution of the original data (at best, it can preserve the resolution), and no m/z remainder approach can improve the mass accuracy of the original data. The true test should be done by utilizing the subset of real high-resolution data obtained on TOF

or Orbitrap instruments, and compare the performance. Additionally, Supplementary Figure 1 is suspect, as it only considers glycosidic bond cleavages. Cross-ring cleavages and neutral losses are common in glycan tandem mass spectra. Figure S1 does not reflect the true extent of potential occurrence of multiple fragments in a single bin.

This is an important point and we rephrased the statement in our revised manuscript to reflect that the m/z remainder approach can, at best, only recover the resolution of the mass spectrometer. It still stands, however, that we currently do not see any improvements in prediction when we increase the effective resolution of the model (not the mass spectrometer) by binning more finely, as that should also make the m/z remainder recovery more robust. If the hypothesis were true that higher-resolution data, on average, lead to more correct annotations in theory, then this effect should already become apparent here, which we do not find in the old Supplementary Table 3 (now Supplementary Table 4). While there could be a theoretical threshold effect, in that performance only increases when reaching a certain value of resolution and not before, this seems like a highly elaborate scenario to the authors.

In the revised manuscript, we have taken up the suggestion of the reviewer and have added a comparison to the Orbitrap data (we caution that we have insufficient data from TOF set-ups to draw representative conclusions yet), to further test the importance of resolution, found in the **new Supplementary Table 6**. We further caution, however, that comparisons will always be flawed to some extent, as, in our experience, annotations from the ion trap data subset are, on average, much better, because they usually are refined with additional exoglycosidase digestions etc. (seen for instance in the best-in-class performance on set-ups using the amaZon ion trap, which all derive from researchers at the Leiden University Medical Center). As we tend to see less of that in the TOF/Orbitrap data, resulting in more structures that are either only partially defined or lack important reporting details including retention time, this constitutes an inextricable confounder. We also would like to take this opportunity to argue that the, even higher, performance on amaZon data (>95% accuracy in the **new Supplementary Table 6**), despite highly diverse and complex O-glycan samples, already indicates that, with higher-quality data, even higher prediction performance can be reached with CandyCrunch than we currently report across all types of data, making us very optimistic and excited about the future potential of this technology.

11. The authors noted that “a model only trained on O-glycans performed worse for predicting O-glycans (Supplementary Table 4; topology: 84% accuracy, structure: 79% accuracy) than the model trained on all classes”, and hypothesized that “this was due to the structure-based loss function we used for training, as well as shared information between spectra of different classes, stemming from shared glycan motifs across classes”. This is an interesting point. How is the motif information incorporated into the training algorithm? Would CandyCrunch be able to predict structures that are built from existing motif but not in the training set? Again, this could be tested by removing certain structures entirely from the training set.

We agree with the reviewer and think this is very promising. Our structure-based loss function works such that two glycans receive a higher value (a greater distance) if they share fewer substructures. As the model tries to minimize the loss, it is guided toward predicted structures that have similar substructures or motifs. Our software library glycowork has potent capabilities to recognize and count motifs in glycan sequences and we use this functionality to construct this initial structure-based distance matrix for the glycans in our training data. This matrix is then used as a kind of lookup table during training. As discussed in more depth above, CandyCrunch (ignoring biosynthetic modeling etc.) is inherently incapable of predicting unseen glycans, no matter how similar to seen glycans. However, this shared information that is being considered during training allows us to perform better on glycans for which we have few training spectra, if that glycan has many shared motifs with other glycans, for which we do have more training spectra. We tested this cross-training hypothesis by analyzing the dependence of glycan performance on number of training spectra, in the context of whether its motifs are found in other glycans in the data, shown in the **new Supplementary Fig. 6**. This analysis indeed confirms that, with statistical significance, a higher support (i.e., a greater number of spectra from structurally related glycans in the training set) lowers the probability of low-accuracy predictions for glycans with few training spectra on their own.

12. CandyCrumbs is a useful tool, but the way it prioritizes assignments when multiple assignments are possible is suspect (Figure S10). There are often cases where a neutral loss or double cleavages are more abundant than a single-cleavage fragment. Is it possible to also learn the assignment from the annotated training set?

We agree with the reviewer that, at best, our fragment prioritization is a heuristic. First, we would like to note that fragment prioritization is only an option within CandyCrumbs, controlled by a keyword argument, and our method can readily output all possible fragment assignments at a given m/z value. The trade-offs here are that the agnostic output of all possible fragments might be most useful for expert users but not for intermediate users, who might be, for instance, overwhelmed by the inclusion of, theoretically possible, triple-cleavage fragments. This is why we support both modes of analysis. We note that, while there are certainly cases where our heuristic breaks down, the main aim of a heuristic is to be more right than wrong in most of the cases. It also needs to be generalizable. We thus maintain that fragment prioritization, if chosen to be activated, adds value. Yet we, of course, are optimistic that this will be improved in the future.

It is theoretically absolutely possible to learn assignments and we would love nothing more. The main issue here is that there is not enough available data for this, as only a minuscule fraction of deposited glycomics data is accompanied with annotated fragments (and then usually in the form of pictures in supplemental figures). Further, if not accompanied by MS3 data, assignments in the case of multiple options are often not based on rigorous criteria and vary extensively between analysts. We are thus hopeful that, once sufficient data of that type become available, CandyCrumbs can be revisited as a machine learning problem.

Overall, I think the manuscript is worth consideration for publication on Nature Methods, but there are serious flaws that need to be addressed and it needs further and more robust evaluation.

We thank the reviewer for the constructive feedback and we are convinced that our point-by-point responses, as well as the associated changes we made to the revised manuscript, have improved our work substantially.

Reviewer #3:

Remarks to the Author:

The rebuttal addresses all my comments well and the authors updated and improved the manuscript. The described tool has a lot of options to tune it for a specific experiment and train it with new data, and it will likely be very valuable in LC-MS based glycomics experiments in the future.

Although the authors provided an easy accessible Google Drive interface to make the tool accessible for non-python-specialists, this is not an ideal interface to bring the tool to its full potential (as indicated by the authors themselves as well). I understand that this is not the main purpose of the manuscript, but I would like to encourage the developers to equip the Python tool with a user friendly interface to make the tool accessible for all glyco LC-MS specialist, independent of their programming skills. As a start, a clearer “read me” can be provided on how to get started with the tool using the local installation. I’m sure there will be enough “dummy testers” in the field happy to help validating this.

We thank the reviewer for engaging with our work and helping us improve it. As further detailed in the response to Reviewer #5, we have decided to at least also implement a command line interface for using CandyCrunch via simple command line commands (**new v0.3** of CandyCrunch). While we realize that this may still exceed the capabilities or interest of some potential users, we are confident that this still constitutes an improvement over the current state and hope to construct a true graphical interface for this in the future as well. We have also expanded our “Read me” document to be more specific about the usage of CandyCrunch in case of local installation. We are also hopeful that the additions to the revised manuscript (**new Supplementary Fig. 2, new Supplementary Table 12**), detailing usage and options of CandyCrunch, will aid in making this method more accessible.

Reviewer #4:

Remarks to the Author:

This manuscript presents a highly novel tool for glycomics data annotation which is made available via an ad hoc Python notebook. The authors have addressed both the novelties of the software as well as the drawbacks or points for users to be aware of when using the software. In particular, it is very important to make potential users aware of what NOT to expect. The data and methodology is rigorously explained, and the manuscript is written clearly overall.

We are grateful for the evaluation and feedback on our herein described method and address the suggestions for improvement below.

Suggestions for improvement:

1. A table or at least a description of the various options that are available could be summarized. While the author responded to one of the other reviewer's comments and added a statement that the given option (zero-shot) is available, amongst the comments to the reviewers there seemed to be many such options that are available, but the user would not be aware of them. So it would help the manuscript greatly to indicate that all of the various options that often need to be considered when annotation glycomics spectra are addressed, and the list would indicate how to set those options.

We thank the reviewer for this suggestion and have added a schematic description of what needs to be set in using CandyCrunch, and which considerations go into setting parameters (**new Supplementary Fig. 2**), as well as a new supplemental table that describes the options for each of these parameters (**new Supplementary Table 12**). This is also accompanied by a now more extensive "Read me" section on the GitHub page of our CandyCrunch package (**new v0.3** of CandyCrunch).

2. While users could use the outputted annotations as figures to include in their own publications, how do the authors consider feeding back the annotations to databases such as in the GlySpace Alliance? In particular, are GlyTouCan IDs assignable to the annotations?

This is a very valuable suggestion and we have now linked the glycans in our predictions to GlyTouCan IDs, whenever available. Thus, from CandyCrunch **v0.3** onward, GlyTouCan IDs will also be present in the annotation tables of the output of this method and can be used to interface findings with the resources of the GlySpace Alliance.

Reviewer #5:

Remarks to the Author:

The manuscript by Urban et al. describes a method for identification of glycan structures from glycomics mass spectrometry data using deep learning. The method clearly provides a dramatic improvement in capability vs existing software methods, both in performance and generalizability to a variety of experimental and instrumental setups. This is both a very challenging and very necessary endeavor, making the manuscript likely of great interest to a broad glycoscience community. The manuscript is well written and, particularly following the revisions from earlier reviews, clearly describes the method, results, and current limitations. I have some specific comments on the method and results below, but most are very minor issues.

We thank the reviewer for their evaluation of our work and address specific comments below.

The only major issue I see is the requirement that the software be accessed programmatically in Python, with no provision of any user interface or command line accessibility beyond the CoLab notebook demo. While this is not an issue for experienced programmers, to truly “democratize structural glycomics” beyond a few specialists, this represents a major roadblock for glycobiochemists and other glycoscientists who may not have the programming skills (or time) to write their own code to use the method. The authors refer to the CoLab demo as not a main part of the manuscript in responding to the initial review comments about its limited functionality, but these comments are a clear indication of the need and desire for a functional interface to fully realize the potential of the method. This dovetails with concerns about the method receiving long term support and engagement – if the barrier to use is too high and few people adopt the method, there is less feedback and external motivation to keep maintaining and improving it. I would urge the authors to provide at least a minimal command line execution option (e.g., pass in a config file that has the parameters and data paths) to run the analysis and generate at least a basic output table or report. Further development (e.g., of a graphical interface) could come later or as warranted from community feedback, but I think there needs to be some provision to run the software without having to write Python code for it to be considered a community resource as opposed to a specialist tool for bioinformaticians. That said, the CoLab demo and source code clearly show a high level of programming expertise, so I have confidence in the authors’ capability to do this.

We appreciate the feedback on our manuscript and, as we also realize the importance of lowering barriers of entry, we have now added command line execution functionality to give users more flexibility in interacting with our method (**new v0.3** of CandyCrunch). We are also confident that future work might result in a true graphical interface, which would then even further increase its reach and approachability.

Minor Comments:

- “Applied to fully unseen datasets, CandyCrunch routinely achieved high performance (Supplementary Table 4; topology: 92% accuracy, structure: 84%)”. These numbers appear to be Top5 structure accuracies from Supp. Table 4, whereas the previous data in Figure 1 was presented as Top1 accuracy. This needs to be noted, and ideally, the range of Top1 accuracies should be provided as well or instead.

This is correct and we have added this information to the revised manuscript. We now report the corresponding values for Top1 predictions, updated for the re-trained model from the previous round of revisions, as well as improvements in the post-prediction workflow (now from the updated Supplementary Table 8).

- With regard to Supp. Table 3, the authors state that higher m/z resolution appeared not to necessarily translate to higher prediction performance. The resolutions tested (0.7 to 1.4 Da per bin) are 2 orders of magnitude lower than high resolution mass spectrometry data, so it would be worth noting the difference between “higher” resolution than the model default (i.e., 2x higher) and “high” resolution in a mass spectrometry context (i.e., 100x higher).

We thank the reviewer for bringing up this important distinction and have added this information to the revised manuscript. We are also enthusiastic about revisiting this avenue of research once sufficient amounts of high-resolution mass spectrometry data become publicly available.

Decision Letter, second revision:

Dear Daniel,

Thank you for submitting your revised manuscript "Predicting glycan structure from tandem mass spectrometry via deep learning" (NMETH-A52883C). It has now been seen by the original referees and their comments are below. The reviewers find that the paper has improved in revision, and therefore we'll be happy in principle to publish it in Nature Methods, pending minor revisions to satisfy the referees' final requests and to comply with our editorial and formatting guidelines.

TRANSPARENT PEER REVIEW

Please note: we allow redactions to authors' rebuttal and reviewer comments in the interest of confidentiality. If you are concerned about the release of confidential data, please let us know specifically what information you would like to have removed. Please note that we cannot incorporate redactions for any other reasons. Reviewer names will be published in the peer review files if the reviewer signed the comments to authors, or if reviewers explicitly agree to release their name. For more information, please refer to our FAQ page.

ORCID

Sincerely,
Arunima

Arunima Singh, Ph.D.
Senior Editor
Nature Methods

Reviewer #1 (Remarks to the Author):

The updated manuscript represents a substantial improvement over the initial submission. I commend the authors for their diligent efforts in addressing the concerns raised in my original review. They conducted further analyses to assess the impact of various training parameters and/or filtering procedures on the performance of CandyCrunch. In specific cases, they explicitly acknowledged the inherent limitations of their current approach. This acknowledgment is crucial for the audience to appreciate both the strengths and limitations of their methodology. Overall, I am content with the revisions made, but there remain a few outstanding issues that the authors should address or acknowledge to enhance the manuscript further.

(1) In the revised manuscript, the authors acknowledged that zero-shot prediction is only feasible for analyzing mixtures of related glycans. Figure S8A illustrates the application of biosynthetic modeling to recover zero-shot intermediates. The model predicts an O-glycan with Core-5 structure (GalNAc α 1-3(Neu5Gc α 2-6)GalNAc), further confirmed by the near-perfect match of the retention time. However, there might be confusion regarding how the retention time matching is achieved if this glycan structure is removed from the training data. Do authors imply that retention time of a glycan can be predicted based on the retention times of its related structures?

Additionally, were Core 3 (GlcNAc- β 13-GalNAc), Core 6 (GlcNAc- β 1-6-GalNAc), Core 7 (GalNAc- α 16-GalNAc) structures also present in this experimental dataset? If so, how did CandyCrunch rank other potential candidate structures? If not, the authors should take this opportunity to emphasize the benefits of utilizing biosynthetic modeling for zero-shot predictions, especially considering that these

other structures are biologically relevant.

(2) The authors' statement that it is rare to find cases where isomers co-elute within a half-minute retention time window may seem unexpected. For instance, a recent study investigating high-mannose and paucimannose N-glycans (J. Proteome Res. 2024, 23, 939-955) demonstrates significant overlap of isomers, with many falling within a 0.5-minute retention time window (Table 1, noting that 1 dextran index corresponds to approximately a 2–3-minute gap in retention time). However, it should be noted that the JPR study did not involve reduction, resulting in more complex chromatograms due to anomerism.

The authors' statement that the examples shown in Figure S3 are isobars and not strictly isomers is incorrect. Gal β 1- β GalNAc α 1-3(Neu5Ac α 2-6)GalNAc and Fuca1- β GalNAc α 1-3(Neu5Gc α 2-6)GalNAc are indeed isomers, as both should have an m/z value of 878.326 in their reduced and deprotonated form, or 878.325 in their underivatized and protonated form. The difference in their m/z values in the datasets are likely due to insufficient mass measurement accuracies.

(3) The authors stated that they currently do not see any improvements in prediction when they increase the effective resolution of the model by binning more finely, but acknowledged that the performance may only increase when reaching a certain value of resolution and not before. They also stated that utilization of the higher resolution Orbitrap data did not lead to performance enhancement. However, it's important to note that Table S4 shows that the highest effective resolution per bin tested is merely 0.72 Da, which is significantly greater than the difference between common isobars observed in glycan fragments (0.036 Da for the CH₄ and O splitting). Therefore, it is not surprising that the benefits of higher mass resolution cannot be realized in the present setting. The authors also acknowledged in their rebuttal letter that direct comparison between performance on data acquired on different instrument platforms is confounded by the variance in other experimental variables, such as validation from exoglycosidase digestions, potential MS³ data (maybe?), annotation quality, and retention time information. These are all valid points that should be incorporated into the main text too inform the readers, rather than being solely addressed in the rebuttal letter. Mass calibration is another complicating factor that the authors should address in their manuscript. Currently, further reducing the bin size may not lead to the desired performance enhancements due to two main factors: (1) to effectively account for the CH₄-O isobar, the bin size would need to be reduced by more than an order of magnitude, which may not be practical; and (2) the variations in mass accuracy of the training data across different datasets do not justify the effort of further reducing the bin size. Instead of conducting performance evaluation at further refined bin sizes, it may be sufficient for the authors to acknowledge these caveats in the revised manuscript.

Reviewer #3 (Remarks to the Author):

The manuscript and tool are greatly improved and in my opinion ready for publication. I look forward to future developments and more user friendly interfaces.

Reviewer #3 (Remarks on code availability):

I was able to run the code, a read me is present.

Reviewer #4 (Remarks to the Author):

All comments have been addressed as well as they could given the current availability of datasets.

Reviewer #4 (Remarks on code availability):

While the source code for prediction appears complete, model training data is unavailable. Scripts to train the model have been uploaded four months ago, but the data to train the model is not, so the training code cannot be executed without an error. This should certainly be added to allow users to train their own datasets and/or get an idea of what the training data looks like.

Reviewer #5 (Remarks to the Author):

The authors have satisfactorily resolved all previous issues I noted with the manuscript, and I have no further comments. The new command line interface for CandyCrunch is easy to use and satisfies my previous concerns about accessibility.

Reviewer #5 (Remarks on code availability):

The software is open source and available on Github and is well documented and clearly structured. I installed the latest version of the software following the instructions on the Github page and was able to easily access the command line interface as described.

Reviewer #6 (Remarks to the Author):

Co-reviewed with Reviewer #1.

Author Rebuttal, second revision:

Reviewer #1 (Remarks to the Author):

The updated manuscript represents a substantial improvement over the initial submission. I commend the authors for their diligent efforts in addressing the concerns raised in my original review. They conducted further analyses to assess the impact of various training parameters and/or filtering procedures on the performance of CandyCrunch. In specific cases, they explicitly acknowledged the inherent limitations of their current approach. This acknowledgment is crucial for the audience to appreciate both the strengths and limitations of their methodology. Overall, I am content with the revisions made, but there remain a few outstanding issues that the authors should address or acknowledge to enhance the manuscript further.

We thank the reviewer for their efforts in improving our manuscript. Below, we address the remaining individual points.

(1) In the revised manuscript, the authors acknowledged that zero-shot prediction is only feasible for analyzing mixtures of related glycans. Figure S8A illustrates the application of biosynthetic modeling to recover zero-shot intermediates. The model predicts an O-glycan with Core-5 structure (GalNAc α 1-3(Neu5Gc α 2-6)GalNAc), further confirmed by the near-perfect match of the retention time. However, there might be confusion regarding how the retention time matching is achieved if this glycan structure is removed from the training data. Do authors imply that retention time of a glycan can be predicted based on the retention times of its related structures? Additionally, were Core 3 (GlcNAc- β 13-GalNAc), Core 6 (GlcNAc- β 1-6-GalNAc), Core7 (GalNAc- α 16-GalNAc) structures also present in this experimental dataset? If so, how did CandyCrunch rank other potential candidate structures? If not, the authors should take this opportunity to emphasize the benefits of utilizing biosynthetic modeling for zero-shot predictions, especially considering that these other structures are biologically relevant.

Our zero-shot process using biosynthetic modelling, demonstrated in the indicated Figure S8a, tries to find “matches” for unobserved intermediates by assigning these intermediates to currently unexplained peaks with (i) the correct precursor m/z and that (ii) pass all quality filters (e.g., not too few fragments, glycan fragments, diagnostic fragments for recognized components such as Neu5Gc in this case). In this case, only one peak (the one at the correct retention time) fulfilled all these criteria, achieving the close match in retention time. In this dataset, Core 3 (but not Core 6 or 7) structures were indeed present. Thus, the isomer GlcNAc(b1-3)[Neu5Gc(a2-6)]GalNAc is also found in at least some files. One important point here is that GalNAc(a1-3)[Neu5Gc(a2-6)]GalNAc was always 10-20x as abundant as the GlcNAc-isomer, in all files in which it was present. In this case, the erroneous CandyCrunch prediction of GlcNAc(b1-3)[Neu5Gc(a2-6)]GalNAc at the Core 5 “slot” was removed by our prediction filters, due to too low

prediction confidence, leaving the “slot” open for our biosynthetic network imputation. We have added some of this information to the revised figure legend of Figure S8a.

(2) The authors' statement that it is rare to find cases where isomers co-elute within a half-minute retention time window may seem unexpected. For instance, a recent study investigating high-mannose and paucimannose N-glycans (*J. Proteome Res.* 2024, 23, 939-955) demonstrates significant overlap of isomers, with many falling within a 0.5-minute retention time window (Table 1, noting that 1 dextran index corresponds to approximately a 2–3-minute gap in retention time). However, it should be noted that the JPR study did not involve reduction, resulting in more complex chromatograms due to anomerism.

We agree that, in biological settings, and especially with *N*-glycans, co-eluting isomers are more common than in our data. The reason for their paucity in our data likely lies in the fact that human annotators, on average, tend to not distinguish these isomers in their published annotations (always with the caveat that these statements refer to the datasets on GlycoPOST and other public sources of raw glycomics data), such that for instance type 1 and type 2 LacNAc isomers, which would co-elute very closely indeed, are either typed “Gal(b1-?)GlcNAc” or simply assumed to be the more common “Gal(b1-4)GlcNAc”. In the revised manuscript, we have added this cautionary note to better contextualize this statement.

(3) The authors' statement that the examples shown in Figure S3 are isobars and not strictly isomers is incorrect. Gal?1-?GalNAc α 1-3(Neu5Ac α 2-6)GalNAc and Fuca1-?GalNAc α 1-3(Neu5Gc α 2-6)GalNAc are indeed isomers, as both should have an *m/z* value of 878.326 in their reduced and deprotonated form, or 878.325 in their underivatized and protonated form. The difference in their *m/z* values in the datasets are likely due to insufficient mass measurement accuracies.

We have corrected this oversight in the revised figure legend of Figure S3.

(4) The authors stated that they currently do not see any improvements in prediction when they increase the effective resolution of the model by binning more finely, but acknowledged that the performance may only increase when reaching a certain value of resolution and not before. They also stated that utilization of the higher resolution Orbitrap data did not lead to performance enhancement. However, it's important to note that Table S4 shows that the highest effective resolution per bin tested is merely 0.72 Da, which is significantly greater than the difference between common isobars observed in glycan fragments (0.036 Da for the CH₄ and O splitting). Therefore, it is not surprising that the benefits of higher mass resolution cannot be realized in the present setting. The authors also acknowledged in their rebuttal letter that direct comparison between performance on data acquired on different instrument platforms is confounded by the

variance in other experimental variables, such as validation from exoglycosidase digestions, potential MS3 data (maybe?), annotation quality, and retention time information. These are all valid points that should be incorporated into the main text too inform the readers, rather than being solely addressed in the rebuttal letter. Mass calibration is another complicating factor that the authors should address in their manuscript. Currently, further reducing the bin size may not lead to the desired performance enhancements due to two main factors: (1) to effectively account for the CH4-O isobar, the bin size would need to be reduced by more than an order of magnitude, which may not be practical; and (2) the variations in mass accuracy of the training data across different datasets do not justify the effort of further reducing the bin size. Instead of conducting performance evaluation at further refined bin sizes, it may be sufficient for the authors to acknowledge these caveats in the revised manuscript.

We agree with the reviewer and have amended our revised manuscript to better reflect the current state of not being effectively capable of leveraging the high mass accuracy of current mass spectrometers, as well as the heterogeneity in experimental set-ups and the concomitant differences in data quality.

Reviewer #3 (Remarks to the Author):

The manuscript and tool are greatly improved and in my opinion ready for publication. I look forward to future developments and more user friendly interfaces.

We thank the reviewer for their comments and feedback.

Reviewer #3 (Remarks on code availability):

I was able to run the code, a read me is present.

Reviewer #4 (Remarks to the Author):

All comments have been addressed as well as they could given the current availability of datasets.

Reviewer #4 (Remarks on code availability):

While the source code for prediction appears complete, model training data is unavailable. Scripts to train the model have been uploaded four months ago, but the data to train the model is not, so the training code cannot be executed without an error. This

should certainly be added to allow users to train their own datasets and/or get an idea of what the training data looks like.

We thank the reviewer for pointing this out. While the training data has always been publicly accessible from <https://zenodo.org/records/10029271>, we did not include this information in the training script specifically. We have ensured that this information is prominently displayed in the manuscript & will amend that in the training script in the upcoming 0.4.0 update to our CandyCrunch package (that ensures full compatibility of CandyCrunch with glycowork 1.2.0).

Reviewer #5 (Remarks to the Author):

The authors have satisfactorily resolved all previous issues I noted with the manuscript, and I have no further comments. The new command line interface for CandyCrunch is easy to use and satisfies my previous concerns about accessibility.

We thank the reviewer for helping us improve our work.

Reviewer #5 (Remarks on code availability):

The software is open source and available on Github and is well documented and clearly structured. I installed the latest version of the software following the instructions on the Github page and was able to easily access the command line interface as described.

Reviewer #6 (Remarks to the Author):

Co-reviewed with Reviewer #1.

We also thank this reviewer for their time and effort.

Final Decision Letter:

Dear Daniel,

I am pleased to inform you that your Article, "Predicting glycan structure from tandem mass spectrometry via deep learning", has now been accepted for publication in Nature Methods. The received and accepted dates will be June 13, 2023 and May 17, 2024. This note is intended to let you know what to expect from us over the next month or so, and to let you know where to address any further questions.

Over the next few weeks, your paper will be copyedited to ensure that it conforms to Nature Methods style. Once your paper is typeset, you will receive an email with a link to choose the appropriate publishing options for your paper and our Author Services team will be in touch regarding any additional information that may be required. It is extremely important that you let us know now whether you will be difficult to contact over the next month. If this is the case, we ask that you send us the contact information (email, phone and fax) of someone who will be able to check the proofs and deal with any last-minute problems.

Please note that *Nature Methods* is a Transformative Journal (TJ). Authors may publish their research with us through the traditional subscription access route or make their paper immediately open access through payment of an article-processing charge (APC). Authors will not be required to make a final decision about access to their article until it has been accepted. Find out more about Transformative Journals

You may wish to make your media relations office aware of your accepted publication, in case they consider it appropriate to organize some internal or external publicity. Once your paper has been scheduled you will receive an email confirming the publication details. This is normally 3-4 working days in advance of publication. If you need additional notice of the date and time of publication,

please let the production team know when you receive the proof of your article to ensure there is sufficient time to coordinate. Further information on our embargo policies can be found here: <https://www.nature.com/authors/policies/embargo.html>

If you are active on Twitter/X, please e-mail me your and your coauthors' handles so that we may tag you when the paper is published.

Best regards,
Arunima

Arunima Singh, Ph.D.
Senior Editor
Nature Methods